# LESS IS MORE: ON THE FEATURE REDUNDANCY OF PRETRAINED MODELS WHEN TRANSFERRING TO FEW-SHOT TASKS

## ABSTRACT

Transferring a pretrained model to a downstream task can be as easy as conducting linear probing with target data, that is, training a linear classifier upon frozen features extracted from the pretrained model. As there may exist significant gaps between pretraining and downstream datasets, one may ask whether all dimensions of the pretrained features are useful for a given downstream task. We show that, for linear probing, the pretrained features can be extremely redundant when the downstream data is scarce, or few-shot. For some cases such as 5-way 1-shot tasks, using only 1% of the most important feature dimensions is able to recover the performance achieved by using the full representation. Interestingly, most dimensions are redundant only under few-shot settings and gradually become useful when the number of shots increases, suggesting that feature redundancy may be the key to characterizing the "few-shot" nature of few-shot transfer problems. We give a theoretical understanding of this phenomenon and show how dimensions with high variance and small distance between class centroids can serve as confounding factors that severely disturb classification results under few-shot settings. As an attempt at solving this problem, we find that the redundant features are difficult to identify accurately with a small number of training samples, but we can instead adjust feature magnitude with a soft mask based on estimated feature importance. We show that this method can generally improve few-shot transfer performance across various pretrained models and downstream datasets.

## 1 INTRODUCTION

Large-scale pretraining has proven to be beneficial for downstream tasks, especially under few-shot settings—for example, transferring from an ImageNet pretrained model boosts 1-shot accuracy on CIFAR-100 from less than 10% to 30% (Kornblith et al., 2019). A simple method for transferring a pretrained model is *linear probing*, a widely used transfer algorithm that trains a linear classifier upon frozen features of downstream images extracted by the pretrained model, which is adopted as the standard method for evaluating representation transferability of pre-trained models in fields like supervised learning (Kornblith et al., 2019; Dosovitskiy et al., 2021), self-supervised learning (Grill et al., 2020; Zbontar et al., 2021; He et al., 2022; Oquab et al., 2023) and multimodal learning (Radford et al., 2021; Goel et al., 2022). While being highly efficient by design, linear probing is also effective in terms of performance and is sometimes on par with full fine-tuning under OOD (Kumar et al., 2022) or few-shot settings (Kornblith et al., 2019; Luo et al., 2023).

As the downstream task can vary a lot, the knowledge needed for solving downstream tasks can be much different from that learned during pretraining (Luo et al., 2023; Panigrahi et al., 2023). Since linear probing does not change the pretrained model and regards it as a feature extractor only, the pretrained knowledge lies in the extracted features entirely. Thus we are curious about the questions:

*How much information in the features extracted by the pretrained model is useful for a given downstream classification task? Does the conclusion change with the downstream dataset size?*

In response, we borrow the concept of feature importance from Luo et al. (2022), use it to rank feature dimensions, and select the most important ones as task-specific features. We find that, perhaps surprisingly, using very few selected feature dimensions, sometimes even less than 1%, performs

comparably to using all dimensions under few-shot settings, and increasing the number of remaining dimensions can further increase performance, then drops back. By counting frequencies of feature dimensions appearing among the top task-specific features from a pool of tasks, we further observe that different tasks usually require different task-specific feature dimensions, even when these tasks are sampled from the same dataset. These phenomena together indicate that the knowledge needed for downstream tasks can vary a lot from task to task, and there usually exists a notable mismatch between knowledge that resides in the pretrained features and the knowledge needed for downstream tasks, at least under few-shot settings, suggesting the importance of feature selection.

By varying the downstream dataset size, we find that, for linear probing, the feature redundancy phenomenon only exists under few-shot settings and nearly all feature dimensions become useful as we increase the number of shots. Thus this phenomenon may serve as an indicator that can characterize the "few-shot" nature of few-shot transfer problems, which is the thing that most previous works failed to figure out but is important for understanding the unique difficulties in few-shot transfer learning compared to many-shot transfer learning, especially for large-scale pretrained models for which overfitting to few-shot samples seems not to be a problem anymore (Luo et al., 2023).

To better understand the feature redundancy phenomenon of pretrained models, we give an example where features only have two dimensions and are sampled from a Gaussian distribution. While being simple, our example reproduces the feature redundancy phenomenon perfectly. By analyzing the Gaussian model theoretically, we find that dimensions with small feature importance, that is, having high variance and a small distance between class centroids, can potentially bias the classification boundary. This combined with the high variance of test samples, negatively influences classification under few-shot settings. When the number of shots increases, linear probing can identify these confounding dimensions and rectify the classification boundary appropriately. Any dimension thus becomes useful as more data comes in, as long as it carries new discriminative information.

Finally, we give an initial attempt at dealing with the feature redundancy phenomenon. We encounter the dilemma of being unable to accurately locate redundant features with a small number of samples. As an alternative solution, we find that adjusting feature magnitude with a soft mask based on feature importance estimated by augmented data works well under few-shot setting for various pretrained models and downstream datasets.

**Main Findings and Contributions.** (1) We discover the feature redundancy phenomenon of pretrained models when transferring to downstream few-shot tasks with linear probing. (2) We give a theoretical analysis of the phenomenon which explains why it exists and why it only happens under few-shot settings. (3) We reveal the difficulty of identifying redundant dimensions with a few samples, and propose to instead use soft mask based on feature importance estimated by augmented data which works well for most few-shot transfer scenarios.

## 2 RELATED WORK

With ever-larger datasets (Deng et al., 2009; Sun et al., 2017; Radford et al., 2021), architectures (Dosovitskiy et al., 2021; Abnar et al., 2022; Dehghani et al., 2023) and improved algorithms for pretraining, vision models now have incredible transfer performance on a range of downstream tasks. This trend started from supervised pretraining several years ago, where a remarkable work (Kornblith et al., 2019) verified that models transferred from supervised ImageNet models generally perform much better than those trained from scratch on downstream tasks, especially under few-shot settings. Self-supervised pretraining then caught attention due to its superior transferability compared to supervised counterparts (Zhao et al., 2021; Islam et al., 2021; Ericsson et al., 2021), and was scaled up to hundreds of millions recently (Oquab et al., 2023). Multimodal pretraining like CLIP (Radford et al., 2021) utilizes hundreds of millions of image-text pair for pretraining, and achieves very promising zero/few-shot transfer performance. In our paper, we show that the feature redundancy phenomenon exists generally in pretrained models regardless of the scale and paradigm of pretraining.

To our best knowledge, little is known about how redundant the features of pretrained models are. One relevant work (Raghu et al., 2017) shows that linearly projecting the features pretrained on CIFAR-10 into a small-dimensional subspace ($< 5\%$) provides nearly the same test accuracy as using the original features in image classification tasks. While this shows the pretrained features are

somewhat redundant even for in-distribution tasks, the redundant features they found are *not* harmful, different from our findings that most redundant features are harmful when transferring pretrained features to few-shot tasks. In the context of few-shot transfer, Das et al. (2022) consider finetuning only a small number of selected features during transfer and achieve promising performance, but they do not give an understanding of why selecting features can be useful.

The oracle feature importance adopted in our work is originally defined by Luo et al. (2022). The core features identified by this definetion are mostly those with a small average magnitude, which are non-core features defined by Zheng et al. (2023). Both two papers find that pretrained neural networks put wrong emphasis on feature dimensions with a large average magnitude. Our work goes further, showing that most feature dimensions of pretrained models carry disturbing information for few-shot transfer and should be removed entirely.

## 3 PRELIMINARIES

**Task definition.** In transfer learning, we have a pretrained model $f : \mathbb{R}^d \to \mathbb{R}^m$ mapping inputs $x \in \mathbb{R}^d$ to features $z \in \mathbb{R}^m$. A $C$-way downstream classification task $\tau$ contains a training set $D^{tr} = \{(x_i, y_i)\}_{i=1}^N$ and a test set $D^{te} = \{(x_j^*, y_j^*)\}_{j=1}^M$ where $y_i, y_j^* \in [C]$ are the class labels. The task $\tau$ is said to be a $K$-shot task if there are exactly $K$ samples in each class of $D^{tr}$. When $K$ is small, typically less than 20, $\tau$ is said to be a few-shot task. The goal of transfer is to construct a generalizable classifier for task $\tau$, that is, given the pretrained model $f$ and the training set $D^{tr}$, build a classifier $g : \mathbb{R}^d \to \mathbb{R}^C$ (the outputs are logits) that has low test error on the test set $D^{te}$.

**Linear probing and nearest-centroid classifier (NCC) (Snell et al., 2017).** For linear probing, the linear classifier is defined as $g(\cdot) = W f(\cdot) + b$, where $f$ is frozen and $W \in \mathbb{R}^{C \times m}, b \in \mathbb{R}^C$ are the parameters of linear transformation to be learned using methods like logistic regression, gradient descent, etc. NCC can be regarded as a specific linear probing algorithm. Define $p_c = \sum_{i=1}^N [\mathbf{1}_{y_i=c} f(x_i)]$ as the class centroid of class $c$, then the logit of class $c$ of NCC is defined as $g(\cdot)_c = -\|f(\cdot) - p_c\|^2$. Regarding NCC as linear probing needs to set parameters of linear transformation as $W_c = 2p_c$ and $b_c = -p_c^T p_c$.

**Feature importance.** Luo et al. (2022) find a way to estimate the degree of relative importance of each feature dimension of a pretrained model wrt a given *binary* downstream classification task. Specifically, let $D_1, D_2$ denote the data distributions of the two downstream classes. For the $k$-th feature dimension, let $\mu_{1k} = \mathbb{E}_{x \sim D_1}[f(x)_k], \mu_{2k} = \mathbb{E}_{x \sim D_2}[f(x)_k]$ denote the means of the feature dimension of the two classes. Similarly, let $\sigma_{1k} = \sqrt{\mathrm{Var}_{x \sim D_1}[f(x)_k]}, \sigma_{2k} = \sqrt{\mathrm{Var}_{x \sim D_2}[f(x)_k]}$ denote the standard variations, and $\omega_k \in \mathbb{R}^+$ denote the degree of relative importance. Then under mild conditions, the value of $\omega_k$ can be estimated as

$$\omega_k = \frac{|\mu_{1k} - \mu_{2k}|}{\sigma_{1k} + \sigma_{2k}}. \tag{1}$$

By making *oracle* feature adjustment, that is, scaling each feature dimension such that the overall mean of feature $\mu_k = (\mu_{1k} + \mu_{2k})/2$ is proportional to $\omega_k$, the importance of features is correctly adjusted and few-shot transfer performance improves. Note that this is not a practical method since the ground-truth means and the standard variations of the whole feature distribution are not known in advance, but it provides a valuable tool for analysis, as we will do in the next section.

## 4 FEATURE REDUNDANCY OF PRETRAINED MODELS

In this section, we prepare a ResNet-50 model pretrained on ImageNet, and explore how much information in the pretrained features is useful for tasks sampled from downstream datasets. The number of feature dimensions is 2048. Please see appendix E for experiments on more pretrained models and downstream datasets.

**Extreme feature redundancy of pretrained models.** Figure 1(a) shows the singular value decomposition in sorted order on the correlation matrix of pretrained features collected on the Aircraft Dataset (Maji et al., 2013). As seen, most singular values are very close to zero. In fact, the effective rank (Roy & Vetterli, 2007) of the feature space (see details in Appendix D) is $86.40$, only about

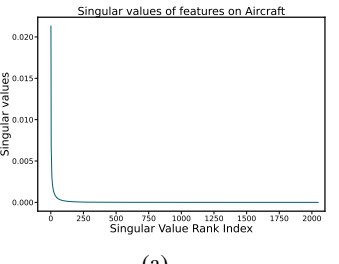
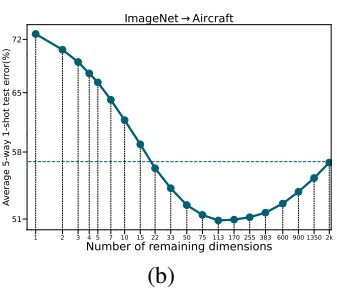
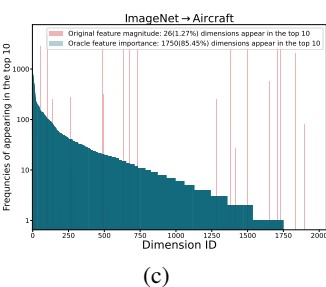

(a)  (b)  (c)

Figure 1: Feature redundancy phenomenon of pretrained models. (a) Singular value spectrum of the feature space of ImageNet-pretrained ResNet-50 computed on AirCraft. (b) Features of pretrained models are highly redundant for few-shot transfer tasks. We mask out unimportant feature dimensions based on feature importance and apply linear probing on the remaining ones. The test error is averaged over 2000 5-way 1-shot tasks sampled from Aircraft. Both axes are logit-scaled. (c) Frequencies of feature dimensions appearing among the top 10 most important features (in blue) or having the largest top 10 average magnitude (in red) on all binary tasks sampled from Aircraft.

$4.3\%$ of the feature dimensionality $2048$. This means that the feature dimensions are highly repetitive and redundant: we can find a very small feature subspace that is equivalent to the original feature space. Are all information contained in this subspace useful for downstream task performance? We select useful dimensions by utilizing feature importance calculated by equation 1. As there may be more than two classes in a practical downstream task, we first generalize equation 1 to make it applicable to multi-class scenarios, simply by averaging the feature importance of all binary tasks constructed by each pair of classes in the downstream task. For a given downstream task, we rank feature dimensions according to their feature importance wrt the task, and then gradually decrease the information contained in the features by masking out unimportant dimensions. We show the 5-way 1-shot performance In Figure 1(b). We can observe that, as we remove feature dimensions gradually (from right to left), few-shot transfer error gradually decreases. This trend continues until the number of remaining dimensions reaches around 110, which is about only 5% of the full dimensions. Notably, using these 5% useful dimensions decreases test error from 56.94% to 50.86%, and using about 1% dimensions can still obtain an accuracy comparable to using the full representation, showing a strong phenomenon of feature redundancy.

**Remark.** The existence of feature redundancy phenomenon indicates that, on the feature level, there exists a notable mismatch between knowledge residing in the pretrained model and the knowledge needed for downstream tasks, at least under few-shot settings. Those feature dimensions that are particularly useful for the task can be seen as *task-specific features* that specialize in the skill needed for the task. This point of view is similar to the observation in Panigrahi et al. (2023), where they found that there exists a small set of *task-specific* parameters ($< 1\%$ of model parameters) responsible for the downstream performance of pretrained models, in the sense that only fine-tuning these parameters gives a performance almost as well as the fully fine-tuned model. The difference is that, task-specific features represent the part of pretrained knowledge that is useful for the current task, while task-specific parameters represent new task knowledge the pretrained model currently lacks and should be learned through fine-tuning. The feature redundancy phenomenon also highlights the importance of feature selection for few-shot transfer, which has been largely neglected in the literature. The difficulty of feature selection lies in the fact that the importance of features is not easy to estimate with only a few examples, which we will discuss further in Section 6.

**Different tasks have different task-specific features.** We now see the importance of task-specific features for downstream few-shot transfer tasks. Then do different tasks have different task-specific features? We calculate the feature importance of all possible binary tasks from Aircraft ($n$ classes produce $n(n-1)/2$ tasks), and visualize the number of times each feature dimension shows up among the top 10 most important dimensions in Figure 1(c). Perhaps surprisingly, more than 85% of dimensions have been among the most important features in at least one task. Such diversity of discriminative features is unexpected since all tasks are sampled from Aircraft—a fine-grained dataset that seems to, intuitively, have only a small set of discriminative features fixed for all tasks. As a comparison, we also visualize the number of times the average feature magnitude for each feature dimension ranks in the top 10 across all feature dimensions in Figure 1(c). Only about

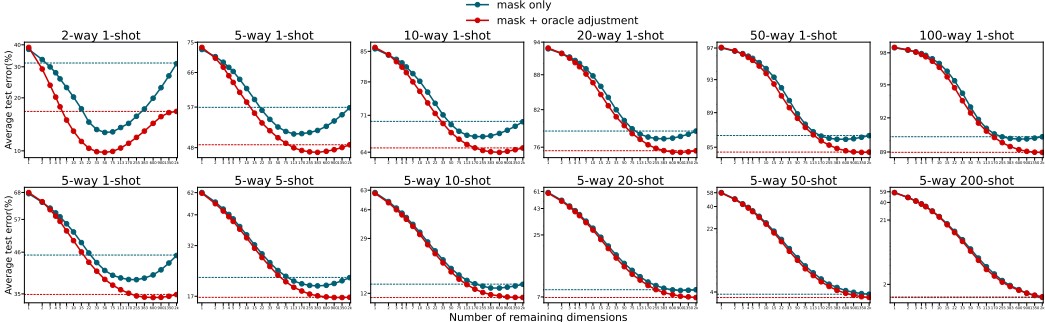

Figure 2: How feature redundancy phenomenon changes with the number of ways and shots. The red curves show the performance obtained by first masking out unimportant features and then adjusting feature importance of the remaining dimensions. We use Traffic Signs (Houben et al., 2013) as the downstream dataset for the second row to have enough shots per class. Both axes are logit-scaled.

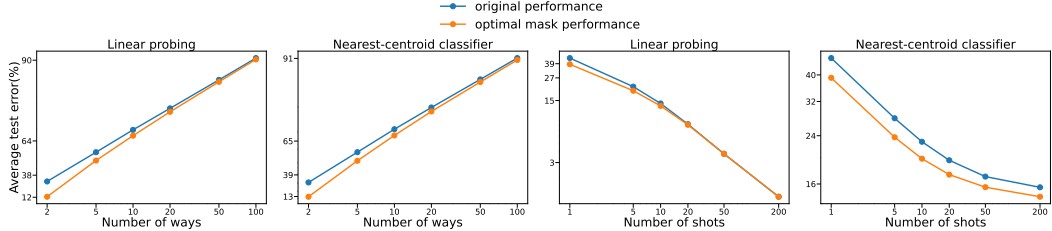

Figure 3: The comparison between the original test error and the error obtained by using the optimal mask when we vary the number of ways and shots. Both axes are logit-scaled. Best viewed in color.

1% feature dimensions have a top-10 large average magnitude in at least one task, and most of these dimensions have low feature importance in most tasks, showing a mismatch between what the pretrained models focus on and what should be focused on.

**Downstream dataset size.** Until now, we have fixed the tasks all to be 5-way 1-shot. Will the feature redundancy phenomenon still exist if we increase the downstream dataset size? We show the full picture of how the feature redundancy phenomenon changes with the number of ways or shots for linear probing in Figure 2, and give the overall trend for linear probing and NCC in Figure 3. In addition to giving results of purely masking unimportant features, we also give the results of first masking unimportant features and then adjusting the magnitude of the remaining feature dimensions (as described in Section 3) in Figure 2. We first note that even when we properly adjust feature magnitude (red curves), there are still redundant dimensions harmful to downstream tasks. Thus these dimensions are truly redundant and need to be removed. As seen from the first row of Figure 2 and the left two plots in Figure 3, the feature redundancy phenomenon weakens as we increase the number of ways. We have a trivial explanation for this from the previous observations in Figure 1(c): as different pairs of classes have different important dimensions, more downstream classes turn more dimensions to be important for the discrimination among some classes ($n$ classes $\rightarrow n + 1$ classes introduce $n$ additional pairs), hence less severe feature redundancy phenomenon.

Now we turn to the second row of Figure 2 and the right two plots in Figure 3. Interestingly, for linear probing, the feature redundancy phenomenon disappears at some point when the number of shots goes to dozens. The case for NCC is different, where the feature redundancy phenomenon weakens at first and then remains almost unchanged after some point around hundreds of shots. What can we inferred from this observation? We note that most current methods for few-shot learning/transfer can in principle work for many-shot transfer problems, or vice versa. They thus do not answer the question: What makes few-shot transfer different from many-shot transfer? Answering this question may help us understand the unique difficulty of few-shot transfer/learning and design problem-specific methods. Our observation reveals that the feature redundancy phenomenon may be the one we need, as it can serve as an indicator for distinguishing few-shot transfer from many-shot transfer and behave differently with different methods, having the potential of serving as a tool to understand the intrinsic scaling properties of downstream algorithms, as we will do in the next section.

Table 1: Quantative evaluation results for the two-dimensional Gaussian example.

| | Linear probing/NCC 1-shot | Linear probing 500-shot | NCC 500-shot |
|---|---|---|---|
| One dimension | **90.83**±0.37 | 95.20±0.010 | **95.21**±0.01 |
| Two dimensions | 75.87±0.68 | **97.36**±0.007 | 84.37±0.02 |

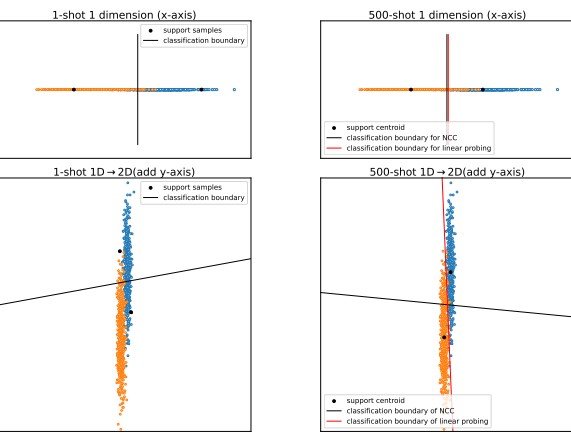

Figure 4: Toy example that illustrates (1) why feature dimensions can be redundant under few-shot settings but turn to be useful when the number of shots increases, and (2) why for NCC the feature redundancy phenomenon does not disappear when the number of shots goes to even hundreds.

## 5  WHAT MAKES FEW-SHOT TRANSFER SPECIAL?

In this section, we provide some theoretical understandings that explain several observations from the previous experiments, including (1) why the feature redundancy phenomenon exists at low-shot settings, (2) why for linear probing, the feature redundancy phenomenon diminishes quickly when we increase the number of shots, and (3) why for NCC, the feature redundancy phenomenon does not disappear when the number of shots goes to even hundreds. Specifically, we consider a two-dimensional Gaussian model for a binary task that is simple for analysis but can still shed light upon the aforementioned questions for complex high-dimensional cases. for label $y \in \{a, b\}$, denote the feature as $z_y$, and we consider the model:

$$z_y \sim \mathcal{N}\left([\mu_{(y,1)}, \mu_{(y,2)}]^T, \mathrm{diag}(\sigma_1^2, \sigma_2^2)\right), \tag{2}$$

where $\mu_{(y,i)}$ is the mean vector of the $i$-th dimension for class $y$, and $\sigma_i^2$ is the feature variance for the $i$-th dimension (we assume the same feature variances for the two classes). To give an intuitive understanding of the feature redundancy phenomenon, we instantiate an illustrative example of this model, with $\mu_{(b,1)} = -\mu_{(a,1)} = 1$, $\mu_{(b,2)} = -\mu_{(a,2)} = 10$, $\sigma_1 = 0.6$, and $\sigma_2 = 10$. This distribution is specially designed, with the feature importance $\omega_1 = 1.67$ of the first dimension much larger than the feature importance $\omega_2 = 1$ of the second dimension. We report few-shot classification performance averaged over 2000 tasks sampled from this distribution using the first or both dimensions in Table 1, and visualize a 1-shot task and a 500-shot task in Figure 4.

**Empirical intuition.** As seen from Table 1 and Figure 4, under 1-shot settings, the first dimension is already discriminative enough for classification: both linear probing and NCC achieve 90.83% accuracy (they are actually the same for 1-shot task). When the second dimension comes in, 1-shot accuracy decreases to 75.87%, indicating that the new dimension is severely redundant and disturbs the classification. We can observe what leads to this result from the lower left plot in Figure 5. As the classifier only sees one training sample per class, the classifier judges a dimension to be informative when the distance between the training samples on this dimension is large. Since for the second dimension, the feature variance $\sigma_2$ is large, the constructed classification boundary is likely to rely more on the second dimension and less on the first dimension. This is problematic because, since the feature variance of the second dimension $\sigma_2$ is large compared to the expected distance between samples $|\mu_{(b,2)} - \mu_{(a,2)}|$ ($\omega_2$ is small), the positions of training samples and test samples

on the second dimension will be both highly unstable, which means that the classification boundary will become highly unstable and the test samples are very likely to distribute across the boundary, leading to decreased performance.

From this point of view, it can be easier to understand why in Table 1, NCC still meets the feature redundancy problem with 500 samples per training class, but linear probing does not. As seen from Figure 4, we can regard a 500-shot classification using NCC as a 1-shot problem where the two well-estimated class centroids have much smaller variances. This avoids the problem of having a too-large distance between training samples on the y-axis caused by high variance, so we can see that the problem is alleviated to some degree. However, if the difference between the means of each class on the y-axis is considerably large, as in the case of our example, the classification boundary will still be somewhat biased to the y-axis; moreover, although the variance of class centroids is small, the variance of the test samples does not decrease, so they are still very likely to pass through the classification boundary when the feature variance of the second dimension $\sigma_2$ is very large compared to the expected distance between samples $|\mu_{(b,2)} - \mu_{(a,2)}|$ ($\omega_2$ is small). So regardless of the number of shots, NCC will meet feature redundancy problem when the feature importance of the second dimension is considerably smaller than that of the first dimension. Unlike NCC, linear probing has a full sense of all training samples and does its best to separate different classes of samples. In addition to estimating means of features, linear probing can estimate the variance of features better as more training data comes in. This leads to an unbiased classifier when the number of training samples goes to infinity. This explains why in Table 1 the 500-shot performance using both dimensions is better than that using a single dimension—any dimension can be useful for linear probing under high-shot settings as long as it carries new discriminative information. In our Gaussian data example where the two dimensions are independent, a better-than-chance discriminative ability of a dimension suffices to give new information, and the second dimension clearly satisfies this condition.

**Theoretical verification.** We give precise theorems for the intuitions shown above. For a task sampled from equation 2 and $y \in \{a, b\}$, $i \in \{1, 2\}$, denote $p_{(y,i)}$ the $i$-th dimension component of the centroid of training samples of class $y$, and $z_i$ the $i$-th dimension component of a random test sample from class $a$, then we have

**Theorem 5.1** (Existence of feature redundancy for NCC and 1-shot linear probing). *Let $n$ be the number of shots. Suppose that $\frac{|\mu_{(a,2)} - \mu_{(b,2)}|}{\sigma_2} > \frac{2.4}{\sqrt{n}}$ and $\frac{|\mu_{(a,1)} - \mu_{(b,1)}|}{\sigma_1} > 2\frac{|\mu_{(a,2)} - \mu_{(b,2)}|}{\sigma_2} + \frac{5.4}{\sqrt{n}}$, then with probability at least $0.9$ over the random draw of the training set, it holds that*

$$\Pr\left[\left\|(z_1, z_2) - (p_{(a,1)}, p_{(a,2)})\right\|_2 > \left\|(z_1, z_2) - (p_{(b,1)}, p_{(b,2)})\right\|_2\right]$$
$$> \Pr\left[\left|z_1 - p_{(a,1)}\right| > \left|z_1 - p_{(b,1)}\right|\right]. \tag{3}$$

Theorem 5.1 basically states that, if we use NCC as the classifier, then regardless of the number of shots, adding a new feature dimension will very likely lead to decreased performance as long as the new dimension has a relatively smaller feature importance defined in equation 1 compared to other feature dimensions, thus answers our first and third questions raised at the beginning of this section. The theorem also justifies our dimension ranking criterion based on the magnitude of feature importance in Section 4. For linear probing, we have

**Theorem 5.2** (Benefit of more feature dimensions for Linear Probing with more shots). *Let $n$ be the number of shots, then let $\hat{h}_{1,n}$ and $\hat{h}_{2,n}$ be the optimal empirical linear classifiers learned by linear probing using only the first dimension and all dimensions respectively. Suppose that $n > 4$, $9\sqrt{\log n / n} < 1$ (to avoid trivial bound), and the mean and variance of data are in the constant order, then with probability at least $0.9$ over the random draw of the training set, it holds that*

$$\Pr\left[\hat{h}_{2,n}([z_1, z_2]) \neq y\right] - \Pr\left[\hat{h}_{1,n}(z_1) \neq y\right]$$
$$\leq \underbrace{\Phi\left(\frac{|\mu_{(a,1)} - \mu_{(b,1)}|}{2\sigma_1}\right) - \Phi\left(\sqrt{\frac{(\mu_{(a,1)} - \mu_{(b,1)})^2}{4\sigma_1^2} + \frac{(\mu_{(a,2)} - \mu_{(b,2)})^2}{4\sigma_2^2}}\right)}_{\leq 0} + 9\sqrt{\frac{\log n}{n}},$$

where $\Phi(\cdot)$ is the cumulative distribution function of the standard Gaussian distribution which is monotonically increasing. Theorem 5.2 basically states that, for linear probing, adding a new feature dimension will more likely lead to improved performance when the number of shots $n$ increases, i.e.,

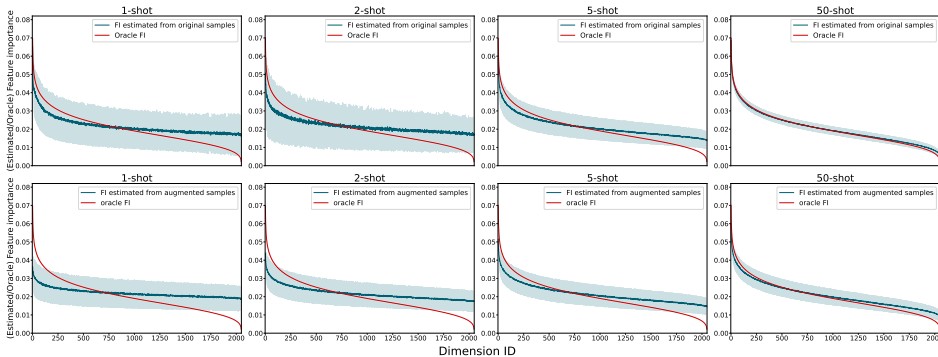

Figure 5: Visualization of how well the feature importance (FI) estimated from few samples approximates the oracle FI. In the first row, FI is estimated using original training samples, while in the second, FI is estimated using samples augmented from the original samples. In each plot, we sample 2000 tasks, and each dimension ID represents the dimension that has the $i$-th largest oracle FI in each task. The curves show the average estimated FI and oracle of all these dimensions, and the bands give standard deviation. Best viewed in color.

the upper bound becomes negative. As we show in the proof in Appendix B, the intuition comes from the fact that the optimal Bayes classifier can make the best use of the added dimension as long as it brings new information, and we prove that the optimal empirical linear classifiers will converge to the optimal Bayes classifier as the number of shots goes to infinity, thus letting the newly added dimension more likely to be useful.

## 6 ATTEMPTS AT ESTIMATING FEATURE IMPORTANCE WITH FEW SAMPLES

In previous sections, feature importance is used as a tool for analysis only. As we have shown, masking out unimportant feature dimensions or making oracle adjustments to the features using feature importance (FI) can bring considerable few-shot transfer performance improvement. However, the calculation of feature importance relies on the class means and variances of the global data distribution, which cannot be obtained in realistic few-shot problems. We are thus interested in whether we can estimate the feature importance using unbiased estimators of class means and variances with the training data of the downstream task. For the 1-shot setting, we cannot estimate the variance, thus we just set all class variances of all dimensions to a fixed value.

**Difficulty of estimating feature importance.** We show in the first row of Figure 5 how well the estimated FI approximates the oracle FI with different numbers of shots. In each plot, the $i$-th dimension ID represents the dimension that has the $i$-th largest oracle FI in each of 2000 sampled tasks. We average all estimated and oracle FIs of these dimensions in each task, and show the results with standard deviation. The distance of the blue lines to the red lines and the variance of two lines in each plot measures how well the estimated FI approximates the oracle FI. We can first observe that, there are a small number of dimensions for each task that have a very large or small oracle FI. As we have analyzed in the previous sections, these dimensions are particularly important: those dimensions having too small oracle FIs are redundant dimensions that need to be masked out, while those having large oracle FIs are task-specific dimensions that need to be highlighted.

As seen, the estimated FI has a large variance throughout all dimensions under few-shot setting. In particular, we see that the average estimated FI has a large gap with the oracle FI at the head and tail of the dimensions. We blame this phenomenon as the result of the difficulty of determining a small set of special dimensions with a limited number of samples for estimation. This shows the fundamental difficulty of figuring out redundant and task-specific dimensions. We verify this observation in Figure 6, where we show that using the estimated FIs cannot help us figure out the redundant dimensions. Another interesting observation is that the variance of the estimated FIs for 2-shot setting is even larger than that for 1-shot setting. In Figure 6, we also show that the estimated FI for 1-shot setting works better than that for 2-shot setting. We note that for 1-shot setting, we just set all class variances to be a fixed value, which is already a no-prior estimation. Thus the variance estimated by the 2-shot data is extremely biased due to untolerable data variance.

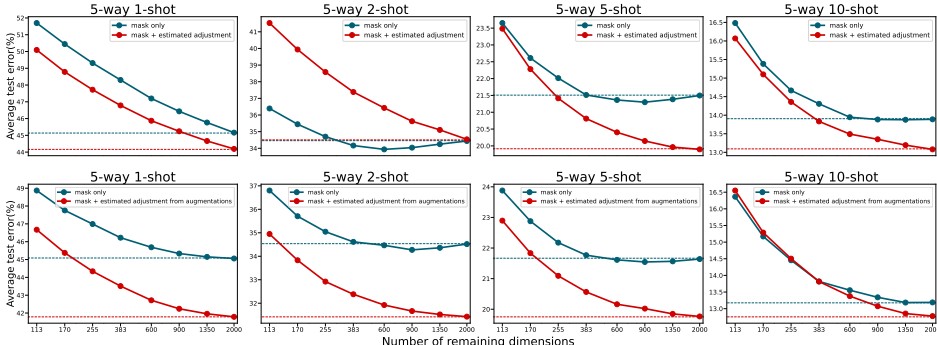

Figure 6: Redundant features cannot be estimated well using few data, but the estimated feature importance can serve as soft masks that help improve few-shot transfer performance. The red curves in the first row show the results of estimation from the original samples, while those in the second row show the results of estimation from augmented samples.

Table 2: 5-way Few-shot transfer Performance improvement brought by adjusting feature magnitude based on the estimation of feature importance from augmented samples, averaged over 2000 tasks.

| | | CUB | Traffic | Aircraft | CropD | ESAT | Fungi | ISIC | Omniglot | QuickD | Flowers | ChestX |
|---|---|---|---|---|---|---|---|---|---|---|---|---|
| DINOv2 | 1-shot | 97.00+0.0 | 51.79+3.0 | 66.50+0.0 | 90.08+0.1 | 64.68+1.5 | 70.83+0.5 | 31.06+0.6 | 81.76+0.4 | 68.26+0.0 | 99.79+0.0 | 22.28+0.0 |
| | 5-shot | 98.84+0.0 | 73.17+3.1 | 74.70+1.0 | 95.63+0.3 | 80.42+2.2 | 86.62+1.1 | 41.83+1.8 | 93.70+1.2 | 86.23+0.2 | 99.93+0.0 | 25.17+0.2 |
| EsViT-base | 1-shot | 68.10+0.4 | 58.16+1.3 | 33.10+0.8 | 79.04+1.6 | 67.15+2.2 | 52.70+2.0 | 31.70+0.7 | 83.58+2.0 | 52.54+1.4 | 82.89+2.1 | 21.89+0.1 |
| | 5-shot | 84.86+1.6 | 76.47+2.2 | 37.48+7.2 | 92.59+0.8 | 85.64+1.2 | 74.30+3.0 | 44.31+1.1 | 93.81+1.4 | 71.74+1.8 | 96.58+0.6 | 24.87+0.2 |
| Res50ImageNet | 1-shot | 74.93+1.6 | 54.58+2.7 | 40.74+1.9 | 74.39+2.5 | 69.52-1.0 | 50.36+2.7 | 29.58+1.1 | 80.63+1.6 | 50.96+1.2 | 79.17+3.5 | 21.85+0.2 |
| | 5-shot | 90.76+1.2 | 73.68+4.1 | 57.37+3.7 | 89.48+1.8 | 84.71+0.8 | 71.47+4.0 | 40.56+2.3 | 91.45+2.1 | 68.85+2.7 | 93.57+1.6 | 24.75+0.1 |
| CLIP | 1-shot | 85.59+3.0 | 60.51+4.9 | 65.69+1.5 | 77.92+3.5 | 62.64+4.6 | 53.78+4.3 | 30.98+1.2 | 85.79+3.1 | 65.50+2.3 | 92.76+2.2 | 21.29+0.2 |
| | 5-shot | 95.97+0.9 | 78.60+6.2 | 79.29+1.9 | 89.93+3.2 | 77.94+6.6 | 71.64+6.3 | 41.74+3.2 | 94.26+2.6 | 83.18+2.0 | 98.74+0.5 | 22.84+0.3 |
| IBOT-base | 1-shot | 74.78+1.5 | 56.41+1.8 | 35.82+0.0 | 83.69+0.1 | 73.11-0.2 | 56.92+0.3 | 33.62+0.4 | 87.32+1.1 | 56.46-0.2 | 87.62+1.3 | 23.23+0.0 |
| | 5-shot | 91.12+1.4 | 76.60+3.0 | 47.23+5.7 | 94.99+0.4 | 89.07+0.3 | 76.72+1.5 | 47.72+0.9 | 95.84+1.2 | 73.95+1.4 | 97.67+0.4 | 27.15+0.3 |
| Swin-base | 1-shot | 77.54+1.2 | 52.53+2.0 | 42.50+0.6 | 79.30+0.8 | 62.29+2.8 | 52.98+1.3 | 29.26+0.6 | 84.90+0.5 | 57.93+0.3 | 80.68+1.0 | 21.87+0.2 |
| | 5-shot | 90.32+0.6 | 73.31+2.5 | 52.92+3.0 | 92.50+0.6 | 80.65+2.1 | 71.11+2.3 | 39.93+1.6 | 94.26+0.9 | 76.47+1.0 | 94.10+0.6 | 24.45+0.3 |

**Better estimation by data augmentation.** Since the uncertainty induced by having too small data for estimation is the fundamental problem of estimating FIs, a natural solution may be augmenting training data and using the augmented data for estimation. A requirement is that the augmented data should mimic the data distribution well. Since for different datasets, the proper data augmentation can be much different, we only use the widely-used random cropping operation. In the second row of Figure 5, we show the FI estimated by 5 augmented samples indeed have a smaller variance for each dimension. In the second row of Figure 6, we also witness that adjusting features using the FI estimated by augmented samples indeed performs much better. However, the variance still exists, thus we still cannot estimate redundant dimensions well. In addition, we note that when the number of shots increases, there always exists a gap between the estimated FI by augmented samples and the oracle FI. This indicates that the augmented samples do deviate a little from the real data distribution.

We can conclude from Figure 6 that the redundant dimensions cannot be estimated well, but using the estimated FIs to adjust features can work well throughout all number of shots. In fact, we can regard the adjustment as a soft mask for those dimensions we are uncertain about whether we should mask or not to a small magnitude, serving as a surrogate way for unconfident situation where overconfidence can lead to problems. We close this section by showing how adjusting features with estimated FIs can generally improve few-shot transferability of pretrained models using linear probing on a range of datasets in Table 2. See Appendix E for the details of all pretrained models and Appendix F for ablation studies.

## 7 CONCLUSION

This paper uncovers the feature redundancy phenomenon that generally exists in pretrained vision models and gives an initial analysis that leads to a better understanding of the "few-shot" nature of few-shot transfer problems. The possible future directions include a better way of estimating redundant features under few-shot settings as well as investigations of the feature redundancy phenomenon under full fine-tuning.

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

# A  PROOF OF THEOREM 5.1

*Proof.* We first note that for the nearest-centroid classifier, shift operation on all features $z \leftarrow z + a$ does not change the classification results. Thus without loss of generality, we assume that $\mu_{(a,1)} + \mu_{(b,1)} = 0$, $\mu_{(a,2)} + \mu_{(b,2)} = 0$, $\mu_{(a,1)} < 0$, $\mu_{(a,2)} < 0$. For the simplicity of presentation, we define $\mu_1 = \mu_{(b,1)}$, and $\mu_2 = \mu_{(b,2)}$. Then each dimension of the class centroids of the training sets and the test sample satisfy

$$p_{(a,1)} \sim \mathcal{N}\left(-\mu_1, \frac{\sigma_1^2}{n}\right),\ p_{(b,1)} \sim \mathcal{N}\left(\mu_1, \frac{\sigma_1^2}{n}\right),\ z_1 \sim \mathcal{N}\left(-\mu_1, \sigma_1^2\right),$$

$$p_{(a,2)} \sim \mathcal{N}\left(-\mu_2, \frac{\sigma_2^2}{n}\right),\ p_{(b,2)} \sim \mathcal{N}\left(\mu_2, \frac{\sigma_2^2}{n}\right),\ z_2 \sim \mathcal{N}\left(-\mu_2, \sigma_2^2\right). \tag{4}$$

Denote the expected error using the first and both dimensions by $L_S(z_1)$ and $L_S([z_1, z_2])$ respectively, where we omit the dependence on the class centroids of the training sets for simplicity. Then according to the definition of NCC, we have

$$\begin{aligned} L_S(z_1) =& \Pr\left[(z_1 - p_{(a,1)})^2 - (z_1 - p_{(b,1)})^2 > 0\right] \\ =& \Pr\left[(p_{(a,1)} - p_{(b,1)})(p_{(a,1)} + p_{(b,1)} - 2z_1) > 0\right], \end{aligned} \tag{5}$$

and

$$\begin{aligned} L_S([z_1, z_2]) =& \Pr\Big[\underbrace{(z_1 - p_{(a,1)})^2 - (z_1 - p_{(b,1)})^2}_{\text{first dimension}} + \underbrace{(z_2 - p_{(a,2)})^2 - (z_2 - p_{(b,2)})^2}_{\text{second dimension}} > 0\Big] \\ =& \Pr\left[(p_{(a,1)} - p_{(b,1)})(p_{(a,1)} + p_{(b,1)} - 2z_1) + (p_{(a,2)} - p_{(b,2)})(p_{(a,2)} + p_{(b,2)} - 2z_2) > 0\right]. \end{aligned}$$

We observe that the class centroids are far apart with high probability, that is,

$$\Pr\big[p_{(a,1)} < p_{(b,1)} \text{ and } p_{(a,2)} < p_{(b,2)}\big] = \Phi\left(\frac{\sqrt{2n}\mu_1}{\sigma_1}\right)\Phi\left(\frac{\sqrt{2n}\mu_2}{\sigma_2}\right), \tag{6}$$

where $\Phi(\cdot)$ is the cumulative distribution function of the standard Gaussian distribution. Thus we assume $p_{(a,1)} < p_{(b,1)}$ and $p_{(a,2)} < p_{(b,2)}$ hold and consider rolling out this probability at last. With this condition, we can further simplify the errors by transforming the term containing $z_1$ and $z_2$ into a standard Gaussian distribution as follows:

$$\begin{aligned}
L_S(z_1) =& \Pr\big[p_{(a,1)} + p_{(b,1)} - 2z_1 < 0\big] \\
=& \Pr\big[p_{(a,1)} + p_{(b,1)} - (-2\mu_1 + 2\sigma_1 * \epsilon_1) < 0\big] \\
=& \Pr\left[\epsilon_1 > \frac{p_{(a,1)} + p_{(b,1)} + 2\mu_1}{2\sigma_1}\right],
\end{aligned} \tag{7}$$

and

$$\begin{aligned}
L_S([z_1, z_2]) =& \Pr\left[2(p_{(b,1)} - p_{(a,1)})z_1 + 2(p_{(b,2)} - p_{(a,2)})z_2 > p_{(b,1)}^2 - p_{(a,1)}^2 + p_{(b,2)}^2 - p_{(a,2)}^2\right] \\
=& \Pr\left[\epsilon_2 > \frac{2(p_{(b,1)} - p_{(a,1)})\mu_1 + 2(p_{(b,2)} - p_{(a,2)})\mu_2 + p_{(b,1)}^2 - p_{(a,1)}^2 + p_{(b,2)}^2 - p_{(a,2)}^2}{2\sqrt{(p_{(b,1)} - p_{(a,1)})^2\sigma_1^2 + (p_{(b,2)} - p_{(a,2)})^2\sigma_2^2}}\right],
\end{aligned} \tag{8}$$

where $\epsilon_1, \epsilon_2 \sim \mathcal{N}(0, 1)$ are independent with both dimensions of all training class centroids. Thus $L_S(z_1) < L_S([z_1, z_2])$ holds if and only if RHS of equation 7 is larger than RHS of equation 8, i.e.,

$$\frac{p_{(a,1)} + p_{(b,1)} + 2\mu_1}{2\sigma_1} > \frac{2\left(p_{(b,1)} - p_{(a,1)}\right)\mu_1 + 2\left(p_{(b,2)} - p_{(a,2)}\right)\mu_2 + p_{(b,1)}^2 - p_{(a,1)}^2 + p_{(b,2)}^2 - p_{(a,2)}^2}{2\sqrt{\left(p_{(b,1)} - p_{(a,1)}\right)^2\sigma_1^2 + \left(p_{(b,2)} - p_{(a,2)}\right)^2\sigma_2^2}}, \tag{9}$$

which can be further simplified as

$$\frac{p_{(a,2)} + p_{(b,2)} + 2\mu_2}{2\sigma_2\sqrt{1 + \frac{(p_{(a,1)} - p_{(b,1)})^2\sigma_1^2}{(p_{(a,2)} - p_{(b,2)})^2\sigma_2^2}}} < \left(1 - \frac{1}{\sqrt{1 + \frac{(p_{(a,2)} - p_{(b,2)})^2\sigma_2^2}{((p_{(a,1)} - p_{(b,1)})^2\sigma_1^2}}}\right)\frac{p_{(a,1)} + p_{(b,1)} + 2\mu_1}{2\sigma_1}. \tag{10}$$

Define $\alpha = \frac{(p_{(a,2)} - p_{(b,2)})^2\sigma_2^2}{(p_{(a,1)} - p_{(b,1)})^2\sigma_1^2}$, then equation 10 is equivalent with

$$\frac{p_{(a,2)} + p_{(b,2)} + 2\mu_2}{2\sigma_2} < \frac{\sqrt{2 + \alpha + \frac{1}{\alpha}} - 1}{\sqrt{1 + \alpha}} \cdot \frac{p_{(a,1)} + p_{(b,1)} + 2\mu_1}{2\sigma_1}. \tag{11}$$

Note that $\frac{\sqrt{2 + \alpha + \frac{1}{\alpha}} - 1}{\sqrt{1 + \alpha}} > 0.5$ holds for all $\alpha > 0$, thus in order for equation 11 to hold, we only need to have

$$\frac{p_{(a,2)} + p_{(b,2)} + 2\mu_2}{2\sigma_2} < \frac{p_{(a,1)} + p_{(b,1)} + 2\mu_1}{4\sigma_1}. \tag{12}$$

By transforming all random variables into a standard Gaussian again, equation 12 is equivalent with

$$\epsilon_3 < \left(\frac{\mu_1}{\sigma_1} - \frac{2\mu_2}{\sigma_2}\right)\frac{\sqrt{10n}}{5}, \tag{13}$$

where $\epsilon_3 \sim \mathcal{N}(0, 1)$. This inequality holds with probability

$$\Phi\left(\frac{\sqrt{10n}}{5}\left(\frac{\mu_1}{\sigma_1} - \frac{2\mu_2}{\sigma_2}\right)\right).$$

Combine this result with equation 6 and relace $\mu_1, \mu_2$ with $|\mu_{(a,1)} - \mu_{(b,1)}|$ and $|\mu_{(a,2)} - \mu_{(b,2)}|$ respectively, we have

$$\Pr\left[L_S(z_1) < L_S([z_1, z_2])\right] > \Phi\left(\frac{\sqrt{2n}|\mu_{(a,1)} - \mu_{(b,1)}|}{\sigma_1}\right) \Phi\left(\frac{\sqrt{2n}|\mu_{(a,2)} - \mu_{(b,2)}|}{\sigma_2}\right)$$
$$\cdot \Phi\left(\frac{\sqrt{10n}}{5}\left(\frac{|\mu_{(a,1)} - \mu_{(b,1)}|}{\sigma_1} - \frac{2|\mu_{(a,2)} - \mu_{(b,2)}|}{\sigma_2}\right)\right). \quad (14)$$

Setting $\frac{|\mu_{(a,2)} - \mu_{(b,2)}|}{\sigma_2} > \frac{2.4}{\sqrt{n}}$ and $\frac{|\mu_{(a,1)} - \mu_{(b,1)}|}{\sigma_1} > 2\frac{|\mu_{(a,2)} - \mu_{(b,2)}|}{\sigma_2} + \frac{5.4}{\sqrt{n}}$, we are able to complete the proof. □

## B  PROOF OF THEOREM 5.2

*Proof.* Similar to the proof of Theorem A, without loss of generality, we set $\mu_{a,1} = -\mu_1$, $\mu_{b,1} = \mu_1$, $\mu_{a,2} = -\mu_2$, $\mu_{b,2} = -\mu_2$. Note that this will not change our argument as the theoretical results are invariant to a shift transformation. Besides, for any classifier in $\mathbb{R}^1$ and $\mathbb{R}^2$ space, we define

$$L_D(h_1) = \Pr[h_1(z_1) \neq y], \quad L_S(h_1) = \frac{1}{n}\sum_{i=1}^{n}\mathbf{1}[h_1(z_{1,i}) \neq y_i];$$

$$L_D(h_2) = \Pr[h_2([z_1, z_2]) \neq y], \quad L_S(h_2) = \frac{1}{n}\sum_{i=1}^{n}\mathbf{1}[h_2([z_{1,i}, z_{2,i}]) \neq y_i].$$

Then according to the definitions of $\hat{h}_{1,n}$ and $\hat{h}_{2,n}$, we have

$$\hat{h}_{1,n} = \arg\min_{h \in \mathbb{R}^1} L_S(h), \quad \hat{h}_{2,n} = \arg\min_{h \in \mathbb{R}^2} L_S(h).$$

Moreover, the optimal population linear classifiers in 1 and 2 dimensions are denoted as follows:

$$h_1^* = \arg\min_{h \in \mathbb{R}^1} L_D(h), \quad h_2^* = \arg\min_{h \in \mathbb{R}^2} L_D(h).$$

In order to prove the performance gap between $\hat{h}_{1,n}$ and $\hat{h}_{2,n}$ (i.e., the difference between $L_D(\hat{h}_{1,n})$ and $L_D(\hat{h}_{2,n})$), we first conduct the following error decomposition:

$$L_D(\hat{h}_{2,n}) - L_D(\hat{h}_{1,n}) \leq L_D(\hat{h}_{2,n}) - L_D(h_1^*)$$
$$= \underbrace{L_D(\hat{h}_{2,n}) - L_D(h_2^*)}_{I_1} + \underbrace{L_D(h_2^*) - L_D(h_1^*)}_{I_2},$$

where the first inequality holds since $L_D(\hat{h}_{1,n}) \geq L_D(h_1^*)$ (see the definition of $h_1^*$). More specifically, the term $I_1$ refers to the performance gap of $\hat{h}_{2,n}$ compared to the population optimal $h_2^*$ and the term $I_2$ refers to the performance gap between the population optimal classifiers $h_2^*$ and $h_1^*$.

**Upper bound for $I_2$.** We first consider $I_2$, for $L_D(h_1^*)$, it is clear that

$$L_D(h_1^*) = \Pr\left[z > 0 : z \sim \mathcal{N}(-\mu_1, \sigma_1^2)\right] = 1 - \Phi(\mu_1/\sigma_1). \quad (15)$$

For $L_D(h_2^*)$, let $[w_1, w_2]^\top$ be the parameter of the linear classifier in 2-dimensional space, we have

$$L_D(h_2^*) = \min_{w_1, w_2} \Pr\left[z_1 \cdot w_1 + z_2 \cdot w_2 > 0 : z_1 \sim \mathcal{N}(-\mu_1, \sigma_1^2), z_2 \sim \mathcal{N}(-\mu_2, \sigma_2^2)\right]$$
$$= \min_{w_1, w_2} \Pr\left[z > 0 : z \sim \mathcal{N}(-\mu_1 \cdot w_1 - \mu_2 \cdot w_2, w_1^2 \cdot \sigma_1^2 + w_2^2 \cdot \sigma_2^2)\right]$$
$$= \min_{w_1, w_2} 1 - \Phi\left(\frac{\mu_1 \cdot w_1 + \mu_2 \cdot w_2}{\sqrt{w_1^2 \cdot \sigma_1^2 + w_2^2 \cdot \sigma_2^2}}\right).$$

Note that $\Phi(x)$ is strictly increasing, it suffices to maximize $\frac{\mu_1 \cdot w_1 + \mu_2 \cdot w_2}{\sqrt{w_1^2 \cdot \sigma_1^2 + w_2^2 \cdot \sigma_2^2}}$, which gives

$$\max_{w_1,w_2} \frac{\mu_1 \cdot w_1 + \mu_2 \cdot w_2}{\sqrt{w_1^2 \cdot \sigma_1^2 + w_2^2 \cdot \sigma_2^2}} = \max_{\alpha} \frac{\mu_1 + \mu_2 \cdot \alpha}{\sqrt{\sigma_1^2 + \alpha^2 \cdot \sigma_2^2}} \geq \sqrt{\frac{\mu_1^2}{\sigma_1^2} + \frac{\mu_2^2}{\sigma_2^2}},$$

where the inequality holds by picking $\alpha = \mu_2^2 \sigma_1^2 / (\mu_1^2 \sigma_2^2)$. This further implies that

$$L_D(h_2^*) = 1 - \Phi\left(\sqrt{\mu_1^2/\sigma_1^2 + \mu_2^2/\sigma_2^2}\right). \tag{16}$$

Combining equation 15 and equation 16, we can finally get the bound for $I_2$:

$$I_2 := L_D(h_2^*) - L_D(h_1^*) \leq \Phi(\mu_1/\sigma_1) - \Phi\left(\sqrt{\mu_1^2/\sigma_1^2 + \mu_2^2/\sigma_2^2}\right). \tag{17}$$

**Upper bound for $I_1$.** Now we focus on the term $I_1$. In particular, we have

$$I_1 := L_D(\hat{h}_{2,n}) - L_D(h^*) = L_D(\hat{h}_{2,n}) - L_S(\hat{h}_{2,n}) + L_S(\hat{h}_{2,n}) - L_S(h_2^*) + L_S(h_2^*) - L_D(h_2^*)$$
$$\leq L_D(\hat{h}_{2,n}) - L_S(\hat{h}_{2,n}) + L_S(h_2^*) - L_D(h_2^*),$$

where the second inequality follows from the fact that $L_S(\hat{h}_{2,n}) = \min_{h \in \mathbb{R}^2} L_S(h)$. Moreover, by Hoeffding's inequality, we can further get that with probability at least $1 - \delta$ for any $\delta < 0.2$:

$$L_S(h_2^*) - L_D(h_2^*) \leq \sqrt{\log(1/\delta)/2n}. \tag{18}$$

Regarding the term $L_D(\hat{h}_{2,n}) - L_S(\hat{h}_{2,n})$, we need to consider a cover of the hypothesis class in $\mathbb{R}^2$. Assume that $\epsilon \ll 1$ and $\mu_1/\sigma_1, \mu_2/\sigma_2 = \Omega(1)$, then for for any data pair $z = [z_1, z_2]$ and negative data $z' = [z_1', z_2']$, we have with probability least $1 - \epsilon/2$, it holds that

$$\|z - z'\|_2 \geq |z_1 - z_1'| \geq \frac{\epsilon}{2} \cdot \sqrt{2\pi}\sigma_1 \cdot e^{\mu_1/(2\sigma_1)}.$$

Similarly, we have with probability at least $1 - \epsilon/2$,

$$\|z - z'\|_2 \geq |z_2 - z_2'| \geq \frac{\epsilon}{2} \cdot \sqrt{2\pi}\sigma_2 \cdot e^{\mu_2/(2\sigma_2)}.$$

This implies that with probability at least $1 - \epsilon$, it holds that

$$\|z - z'\|_2 \geq \frac{\epsilon}{4} \cdot \sqrt{2\pi}\sigma_1 \cdot e^{\mu_1/(2\sigma_1)} + \frac{\epsilon}{4} \cdot \sqrt{2\pi}\sigma_2 \cdot e^{\mu_2/(2\sigma_2)}.$$

By union bound, we can get with probability at least $1 - n^2 \epsilon$, for any pair of data, the above holds for any pair of training dataset. Further by the assumption that the mean and variance of data are in the constant order, we have with probability at least $1 - \epsilon$, it holds that $\|z\|_2 \leq |\mu_1| + |\mu_2| + 2\log(n/\epsilon) \cdot (\sigma_1 + \sigma_2)$ for all training data. Then we have for any 2-dimensional linear classifier $h_2$ with parameter $w \in \mathcal{S}^2$, for any $h_2'$ with parameter $w' \in \mathcal{S}^2$ and satisfies

$$\|w' - w\|_2 \leq \frac{\epsilon}{7\log(n/\epsilon)} \leq \epsilon \cdot \frac{\sqrt{2\pi}\sigma_1 \cdot e^{\mu_1/(2\sigma_1)} + \sqrt{2\pi}\sigma_2 \cdot e^{\mu_2/(2\sigma_2)}}{8(|\mu_1| + |\mu_2| + 2\log(n/\epsilon) \cdot (\sigma_1 + \sigma_2))},$$

with probability at least $1 - 2n^2 \epsilon$, we have

$$|L_S(h_2) - L_S(h_2')| \leq \frac{1}{n},$$

since there will be at most one training data being predicted differently in $h_2$ and $h_2'$.

Moreover, for any such $h_2$ and $h_2'$, let $z = [z_1, z_2]^\top$, we can get

$$|L_D(h_2) - L_D(h_2')| \leq \Pr\left[h_2([z_1, z_2]) \neq h_2'(z_1, z_2)\right]$$
$$\leq \Pr[|w^\top z| \leq |(w - w')^\top z|]$$
$$\leq \Pr[|w^\top z| \leq \|w - w'\|_2 \cdot \|z\|_2)]$$
$$\leq \frac{e^{-\sqrt{\mu_1^2 + \mu_2^2}/(2\sqrt{\sigma_1^2 + \sigma_2^2})}}{\sqrt{2\pi(\sigma_1^2 + \sigma_2^2)}} \cdot [|\mu_1| + |\mu_2| + 2\log(1/\epsilon) \cdot (\sigma_1 + \sigma_2)] \cdot \|w - w'\|_2 + \epsilon$$
$$\leq \frac{\epsilon}{3\log(n/\epsilon)\log(1/\epsilon)} + \epsilon,$$

Table 3: Performance improvement brought by the proposed method using NCC with different distance metrcsL Cosine distance, Euclidean distance and Kendall distance. We use the visual encoder of CLIP as the pretrained model.

| | | CUB | Traffic | Aircraft | CropD | ESAT | Fungi | ISIC | Omniglot | QuickD | Flowers | ChestX |
|---|---|---|---|---|---|---|---|---|---|---|---|---|
| Cosine NCC | 1-shot | 83.36+4.8 | 59.20+5.4 | 64.04+2.8 | 76.72+4.2 | 61.86+4.8 | 51.71+5.1 | 30.27+1.4 | 84.38+3.9 | 61.60+5.0 | 91.18+3.7 | 21.08+0.3 |
| | 5-shot | 96.28+0.7 | 78.34+5.5 | 80.48+1.5 | 91.61+2.0 | 79.33+4.7 | 72.21+5.4 | 41.28+2.7 | 94.86+1.8 | 82.89+2.1 | 98.87+0.4 | 22.87+0.2 |
| Euclidean NCC | 1-shot | 83.72+3.2 | 59.53+4.4 | 64.46+0.7 | 76.58+3.3 | 61.78+3.7 | 51.55+3.6 | 30.34+0.9 | 84.57+2.5 | 61.90+1.4 | 91.71+2.5 | 21.07+0.2 |
| | 5-shot | 96.25+0.7 | 78.40+4.8 | 80.32+1.5 | 91.64+2.0 | 78.90+5.0 | 72.27+5.3 | 41.35+2.3 | 94.90+1.5 | 82.91+1.3 | 98.89+0.4 | 22.86+0.1 |
| Kendall NCC | 1-shot | 86.39+1.3 | 62.26+1.6 | 65.97+0.6 | 78.90+1.3 | 64.71+1.2 | 54.21+1.7 | 30.76+0.4 | 86.73+1.1 | 64.93+1.4 | 93.75+0.8 | 21.19+0.1 |
| | 5-shot | 96.74+0.1 | 81.30+1.5 | 81.28+0.3 | 92.91+0.4 | 82.58+0.6 | 74.99+1.6 | 42.30+0.7 | 95.88+0.5 | 84.32+0.5 | 99.16+0.1 | 22.94+0.1 |

where we use the fact that $e^{-x}x \leq 1$ for all $x \geq 0$ and the fact that $\|w - w'\| \leq \epsilon/(7\log(n/\epsilon))$. Therefore, we can conclude that with probability at least $1 - 2n^2\epsilon$, it holds that for any $h_2'$ that is close to $h_2$ within distance $\epsilon/(3\log^2(n/\epsilon)) + \epsilon$, we have

$$L_D(h_2') - L_S(h_2') \leq \frac{1}{n} + \frac{\epsilon}{3\log(n/\epsilon)^2} + \epsilon + L_D(h_2) - L_S(h_2). \tag{19}$$

Then we are able to design a covering set over $\mathcal{S}^2$, denoted by $\mathcal{H}$, with radius $\epsilon/(7\log(n/\epsilon))$ and the covering number $|\mathcal{H}| = 2\pi \cdot 7^2 \cdot [\log(n/\epsilon)/\epsilon]^2$. Applying Hoeffding's inequality and union bound over the covering sets, we can get with probability $1 - \delta$,

$$\sup_{h \in \mathcal{H}} L_D(h) - L_S(h) \leq \sqrt{\frac{\log(|\mathcal{H}|/\delta)}{2n}} \leq \sqrt{\frac{3\log(\log(n/\epsilon)/(\epsilon\delta))}{n}}. \tag{20}$$

Combining equation 18, equation 19, and equation 20, we are able to get with probability at least $1 - 2\delta - 2n^2\epsilon$, the term $I_1$ can be upper bounded by

$$I_1 \leq \sqrt{\frac{3\log(\log(n/\epsilon)/(\epsilon\delta))}{n}} + \frac{1}{n} + \frac{\epsilon}{3\log(n/\epsilon)^2} + \epsilon + \sqrt{\frac{\log(1/\delta)}{2n}}.$$

Therefore, we can set $\epsilon = 1/(40n^2)$ and $\delta = 0.025$. Then with probability at least $0.9$, we have if $n > 4$,

$$I_1 \leq \sqrt{60\frac{\log(n)}{n}} + \sqrt{\frac{\log(40)}{2}} \leq 9\sqrt{\frac{\log(n)}{n}}.$$

Combining with the bound on $I_2$ in equation 17, we are able to complete the proof.

$\square$

## C    EVALUATION DETAILS

Since the pretraining process of some pretrained models uses normalized features, we normalize all features in all experiments. This leads to the use of cosine distance for the NCC classifier, which is commonly-used in the few-shot learning literature (Chen et al., 2019; Triantafillou et al., 2021). For linear probing, we use the implementation of Logistic Regression by Scikit-learn (Pedregosa et al., 2011) with default settings. For experiments in Section 6, we ensure that the improvement is of statistical significance by sampling the same tasks for each evaluation on one dataset. We consider that linear probing will give a no-class-bias solution to the 1-shot task, which corresponds to a standard Gaussian prior to each class of data. In fact, since for 1-shot task, there is no other information in addition to the single data for each class, a good linear classifier with the best performance expectation should place the classification boundary in the middle of data points, perpendicular to the line connecting them, which is exactly the same with NCC. We empirically find that minimizing Logistic Regression loss leads to this solution (same number for Table 1).

## D    CALCULATION OF EFFECTIVE RANK

Effective rank is a real-valued extension of matrix rank, which is widely-used to calculate the effective dimensionality of matrix and analyze the spectral properties of features in neural networks

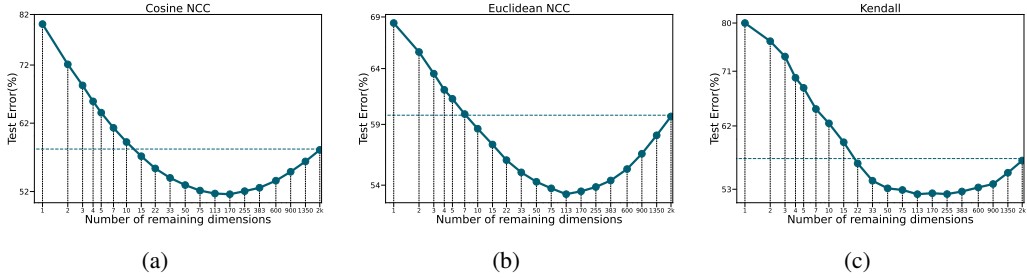

Figure 7: (a) Feature redundancy phenomenon of NCC using Cosine distance. (b) Feature redundancy phenomenon of NCC using Euclidean distance. (c) Feature redundancy phenomenon of NCC using Kendall distance (Zheng et al., 2023). The pretrained model is ResNet-50 trained on ImageNet. The downstream tasks are sampled from Aircraft.

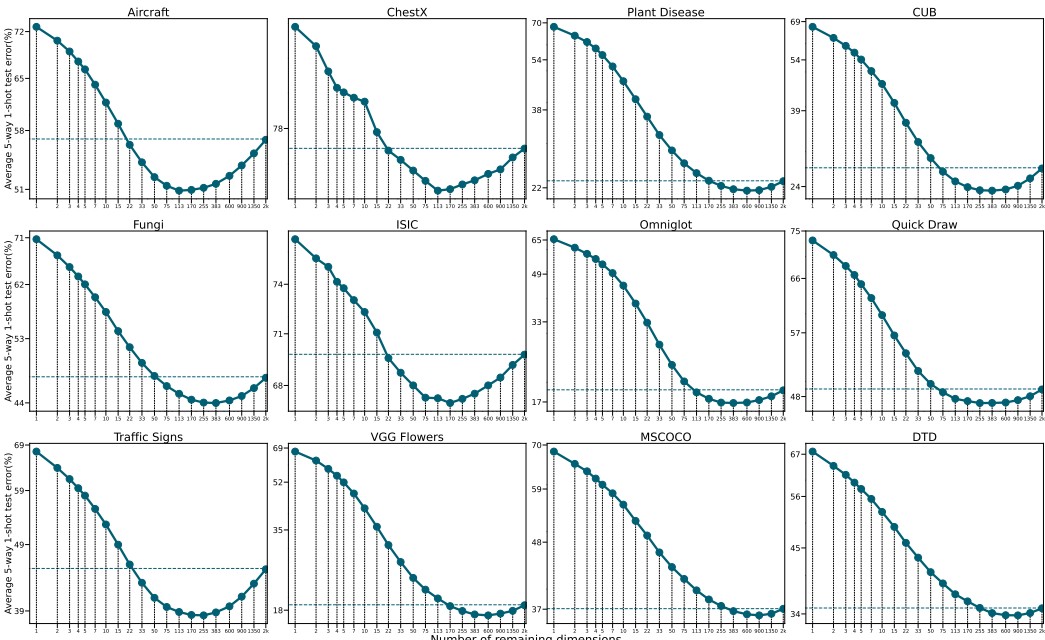

Figure 8: Feature redundancy phenomenon for other datasets. All tasks are 5-way 1-shot.

(Arora et al., 2019; Razin & Cohen, 2020; Huh et al., 2021; Baratin et al., 2021). Given $\{\lambda_i\}_{i=1}^n$ the singular values of a matrix $A$, the effective rank of $A$ is computed as

$$\mathrm{erank}(A) = -\sum_{i=1}^{n} \hat{\lambda}_i \log \hat{\lambda}_i, \tag{21}$$

where $\hat{\lambda}_i = \lambda_i / \sum_k \lambda_k$ is the $i$-th normalized singular value. It has a nice property: $1 \leq \mathrm{erank}(A) \leq \mathrm{rank}(A)$, and $\mathrm{erank}(A) = n$ if and only if $\hat{\lambda}_i = 1/n$ for all $i$.

## E  MORE EXAMPLES OF FEATURE REDUNDANCY PHENOMENON

We give examples of feature redundancy phenomenon of NCC using different distance metrics in Figure 7: Cosine distance, Euclidean distance, and Kendall distance (Zheng et al., 2023), and show the results of our proposed methods for these two distances in Table 3. We can see that, the feature redundancy phenomenon, as an intrinsic problem of pretrained features, generally exists regardless of the transfer-time classifier used.

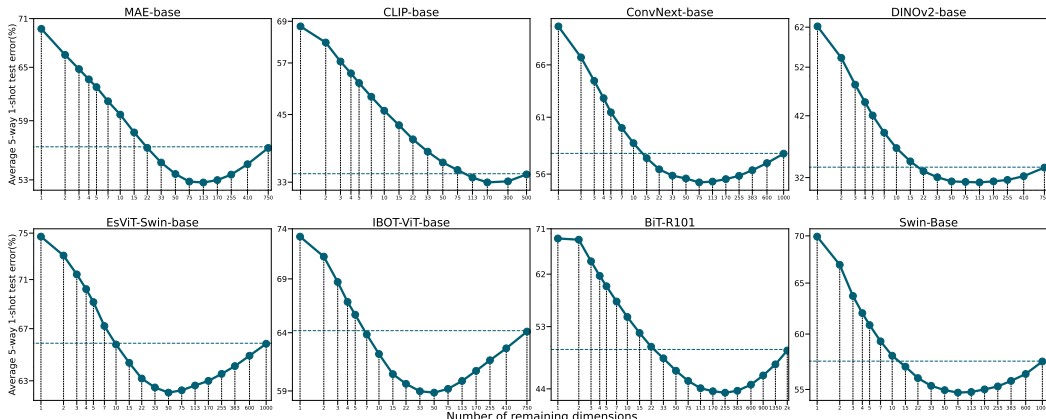

Figure 9: Feature redundancy phenomenon for other pretrained models. All tasks are 5-way 1-shot sampled from Aircraft.

Table 4: 5-way 1-shot ablation studies on the use of augmentations to estimate the feature importance/augment the downstream training set. 'noAug' means we do not use augmentation; 'Aug-est' means we use augmentations to estimate feature importance; 'Aug-est+sample' means we use augmentations to both estimate feature importance and augment the training set of the downstream task (e.g., 5 augmentations per image will alter a 1-shot task into a 5-shot task).

| | CUB | Traffic | Aircraft | CropD | ESAT | Fungi | ISIC | Omniglot | QuickD | Flowers | ChestX | Avg |
|---|---|---|---|---|---|---|---|---|---|---|---|---|
| DINOv2-noAug | 97.00+0.0 | 50.42+0.9 | 66.27+0.0 | 90.01+0.0 | 62.74+0.5 | 70.52+0.1 | 30.56+0.1 | 79.75+1.4 | 65.57+1.5 | 99.77+0.0 | 22.09+0.1 | 66.79+0.4 |
| DINOv2-Aug-est | 97.00+0.0 | 51.79+3.0 | 66.50+0.0 | 90.08+0.1 | 64.68+1.5 | 70.83+0.5 | 31.06+0.6 | 81.76+0.4 | 68.26+0.0 | 99.79+0.0 | 22.28+0.0 | 67.64+0.6 |
| DINOv2-Aug-est+sample | 97.14+0.0 | 53.34+2.1 | 65.83+0.1 | 89.93+0.1 | 60.03+2.1 | 70.77+0.4 | 30.78+0.6 | 73.90+3.3 | 63.54+0.6 | 99.78+0.0 | 22.08+0.0 | 66.10+0.9 |
| EsViT-noAug | 68.47+0.4 | 57.60+0.6 | 33.03+0.3 | 79.19+0.7 | 67.65+0.5 | 52.67+0.4 | 31.67+0.1 | 83.88+0.4 | 52.44+0.6 | 83.25+1.2 | 22.08+0.0 | 57.45+0.5 |
| EsViT-Aug-est | 68.10+0.4 | 58.16+1.3 | 33.10+0.8 | 79.04+1.6 | 67.15+2.2 | 52.70+2.0 | 31.70+0.7 | 83.58+2.0 | 52.54+1.4 | 82.89+2.1 | 21.89+0.1 | 57.35+1.3 |
| EsViT-Aug-est+sample | 68.58+0.2 | 56.32+1.3 | 32.42+1.0 | 77.88+2.1 | 69.32+0.2 | 53.01+1.8 | 31.60+0.6 | 81.68+2.3 | 48.23+2.3 | 82.13+2.2 | 21.69+0.2 | 56.62+1.3 |
| Res50-noAug | 71.70+1.3 | 53.92+1.3 | 41.99+0.1 | 75.76+0.6 | 70.43+0.7 | 50.71+1.4 | 29.88+0.2 | 80.95-0.2 | 49.97+0.3 | 80.07+1.5 | 22.10+0.1 | 57.04+0.7 |
| Res50-Aug-est | 74.93+1.6 | 54.58+2.7 | 40.74+1.9 | 74.39+2.5 | 69.52-1.0 | 50.36+2.7 | 29.58+1.1 | 80.63+1.6 | 50.96+1.2 | 79.17+3.5 | 21.85+0.2 | 56.97+1.6 |
| Res50-Aug-est+sample | 75.54+1.5 | 53.57+3.0 | 40.17+1.5 | 72.76+3.0 | 69.18-1.7 | 51.59+1.9 | 29.52+1.0 | 78.28+2.2 | 45.40+2.7 | 79.03+3.2 | 21.90+0.2 | 56.09+1.7 |
| CLIP-noAug | 83.42+2.5 | 59.26+1.8 | 63.85+2.0 | 77.16+1.4 | 61.46+2.0 | 51.64+2.1 | 30.06+0.7 | 84.63+2.2 | 61.68+3.8 | 91.32+2.4 | 21.09+0.0 | 62.33+1.9 |
| CLIP-Aug-est | 85.59+3.0 | 60.51+4.9 | 65.69+1.5 | 77.92+3.5 | 62.64+4.6 | 53.78+4.3 | 30.98+1.2 | 85.79+3.1 | 65.50+2.3 | 92.76+2.2 | 21.29+0.2 | 63.86+2.8 |
| CLIP-Aug-est+sample | 85.10+3.0 | 57.14+4.5 | 65.48+1.7 | 75.08+4.9 | 58.28+4.5 | 52.53+4.4 | 29.95+1.0 | 78.13+6.2 | 60.49+3.0 | 93.36+1.7 | 21.02+0.2 | 61.51+3.2 |
| IBOT-noAug | 72.99+1.5 | 55.84+0.7 | 34.65+0.3 | 82.20+0.4 | 71.93+0.4 | 54.96+0.4 | 33.30+0.3 | 86.41+1.1 | 54.34+1.2 | 85.68+1.4 | 22.63+0.1 | 59.54+0.7 |
| IBOT-Aug-est | 74.78+1.5 | 56.41+1.8 | 35.82+0.0 | 83.69+0.1 | 73.11-0.2 | 56.92+0.3 | 33.62+0.4 | 87.32+1.1 | 56.46-0.2 | 87.62+1.3 | 23.23+0.0 | 60.82+0.6 |
| IBOT-Aug-est+sample | 73.72+2.1 | 54.85+1.8 | 33.89+0.6 | 81.72+0.8 | 71.41-0.1 | 55.86+0.4 | 33.00+0.5 | 82.73+3.2 | 49.54+2.3 | 86.25+1.8 | 22.73+0.2 | 58.70+1.2 |
| Swin-noAug | 76.52+0.5 | 29.48+0.2 | 41.40+0.3 | 77.55+0.7 | 61.37+0.3 | 50.31+1.2 | 28.79+0.4 | 84.08+0.5 | 55.95+1.1 | 78.25+1.0 | 21.70+0.1 | 55.04+0.6 |
| Swin-Aug-est | 77.54+1.2 | 52.53+2.0 | 42.50+0.6 | 79.30+0.8 | 62.29+2.8 | 52.98+1.3 | 29.26+0.6 | 84.90+0.5 | 57.93+0.3 | 80.68+1.0 | 21.87+0.2 | 58.34+1.0 |
| Swin-Aug-est+sample | 78.43+0.7 | 52.44+1.6 | 41.43+0.9 | 77.72+1.3 | 63.46+2.2 | 52.13+1.4 | 28.94+0.6 | 81.05+1.5 | 51.70+1.8 | 79.48+1.3 | 21.92+0.1 | 57.15+1.2 |

We also give examples of feature redundancy phenomenon on more datasets in Figure 8 and for more pretrained models in Figure 9. The additional datasets include ChestX (Wang et al., 2017), Plant Disease (Mohanty et al., 2016), CUB (Welinder et al., 2010), Fungi (Schroeder & Cui, 2018), ISIC (Codella et al., 2019), Omniglot (Lake et al., 2015), Quick Draw (Jonas et al., 2016), VGG Flowers (Nilsback & Zisserman, 2008), MSCOCO (Lin et al., 2014) and DTD (Cimpoi et al., 2014). The additional pretrained models include supervised ImageNet-1K models SwinTransformer-base (Liu et al., 2021), ConvNext-base (Liu et al., 2022); supervised ImageNet-21K model BiT-R101 (Kolesnikov et al., 2020); self-supervised ImageNet-1K models MAE-base (He et al., 2022), ESViT-Swin-base (Li et al., 2022), IBOT-ViT-base (Zhou et al., 2022); self-supervised model DINOv2-base (Oquab et al., 2023) trained on 142M curated data; and the visual encoder of multimodal model CLIP-base (Radford et al., 2021) trained on 400M image-text pairs. The pretrained models used in Section 6 are the same. We can see that, the feature redundancy phenomenon generally exists regardless of downstream datasets and the choice of pretrained models.

# F    ABLATION STUDIES

In this section, we ablate all considerations of the proposed methods in section 6.

Table 5: 5-way 5-shot ablation studies on the use of augmentations to estimate the feature importance/augment the downstream training set. 'noAug' means we do not use augmentation; 'Aug-est' means we use augmentations to estimate feature importance; 'Aug-est+sample' means we use augmentations to both estimate feature importance and augment the training set of the downstream task (e.g., 5 augmentations per image will alter a 1-shot task into a 5-shot task).

| | CUB | Traffic | Aircraft | CropD | ESAT | Fungi | ISIC | Omniglot | QuickD | Flowers | ChestX | Avg |
|---|---|---|---|---|---|---|---|---|---|---|---|---|
| DINOv2-noAug | 98.95+0.0 | 71.64+2.5 | 73.88+1.8 | 95.77+0.2 | 80.40+1.5 | 86.45+0.9 | 41.37+1.0 | 93.46+1.5 | 85.42+0.4 | 99.94+0.0 | 24.96+0.0 | 77.48+0.9 |
| DINOv2-Aug-est | 98.84+0.0 | 73.17+3.1 | 74.70+1.0 | 95.63+0.3 | 80.42+2.2 | 86.62+1.1 | 41.83+1.8 | 93.70+1.2 | 86.23+0.2 | 99.93+0.0 | 25.17+0.2 | 77.84+1.0 |
| DINOv2-Aug-est+sample | 98.87+0.0 | 69.51+3.7 | 73.45+1.2 | 95.61+0.2 | 74.10+3.2 | 86.32+1.0 | 41.61+1.2 | 89.33+2.3 | 83.45+0.4 | 99.93+0.0 | 24.54+0.2 | 76.07+1.2 |
| EsViT-noAug | 84.31+1.5 | 75.05+2.5 | 37.16+7.2 | 93.08+0.5 | 86.18+0.6 | 73.73+2.5 | 43.52+1.0 | 94.61+1.1 | 72.14+2.0 | 96.44+0.5 | 24.81+0.2 | 71.00+1.8 |
| EsViT-Aug-est | 84.86+1.6 | 76.47+2.2 | 37.48+7.2 | 92.59+0.8 | 85.64+1.2 | 74.30+3.0 | 44.31+1.1 | 93.81+1.4 | 71.74+1.8 | 96.58+0.6 | 24.87+0.2 | 71.15+1.9 |
| EsViT-Aug-est+sample | 84.24+1.6 | 71.34+3.6 | 37.04+4.6 | 91.47+1.2 | 85.55+0.3 | 73.22+3.0 | 43.14+1.0 | 92.52+1.5 | 66.24+2.6 | 96.06+0.7 | 24.48+0.1 | 69.57+1.9 |
| Res50-noAug | 90.14+2.5 | 72.21+5.4 | 59.24+4.3 | 91.50+1.1 | 85.11+0.2 | 71.71+4.3 | 40.94+1.4 | 92.76+1.6 | 68.60+2.9 | 94.41+1.1 | 25.43+0.0 | 72.00+2.3 |
| Res50-Aug-est | 90.76+1.2 | 73.68+4.1 | 57.37+3.7 | 89.48+1.8 | 84.71+0.8 | 71.47+4.0 | 40.56+2.3 | 91.45+2.1 | 68.85+2.7 | 93.57+1.6 | 24.75+0.1 | 71.51+2.2 |
| Res50-Aug-est+sample | 91.27+1.4 | 70.00+5.8 | 55.03+3.8 | 88.86+2.1 | 83.53+0.0 | 71.41+4.2 | 39.82+2.1 | 90.49+2.0 | 62.11+4.3 | 93.37+1.7 | 24.28+0.3 | 70.02+2.5 |
| CLIP-noAug | 96.14+0.7 | 78.18+5.5 | 80.46+1.2 | 91.44+2.0 | 79.45+4.2 | 72.18+4.9 | 41.11+2.1 | 94.89+1.9 | 82.88+2.1 | 98.89+0.3 | 22.81+0.3 | 76.22+2.3 |
| CLIP-Aug-est | 95.97+0.9 | 78.60+6.2 | 79.29+1.9 | 89.93+3.2 | 77.94+6.6 | 71.64+6.3 | 41.74+3.2 | 94.26+2.6 | 83.18+2.0 | 98.74+0.5 | 22.84+0.3 | 75.83+3.0 |
| CLIP-Aug-est+sample | 96.15+0.8 | 74.09+6.6 | 80.49+1.4 | 89.66+3.0 | 71.96+6.9 | 71.05+5.8 | 39.82+2.5 | 90.30+4.6 | 79.64+2.4 | 99.00+0.2 | 22.35+0.3 | 74.05+3.1 |
| IBOT-noAug | 90.69+1.5 | 75.02+3.3 | 46.48+5.7 | 95.15+0.2 | 89.51+0.0 | 76.12+0.9 | 47.45+0.5 | 96.23+1.2 | 74.63+1.5 | 97.48+0.4 | 26.72+0.2 | 74.14+1.4 |
| IBOT-Aug-est | 91.12+1.4 | 76.60+3.0 | 47.23+5.7 | 94.99+0.4 | 89.07+0.3 | 76.72+1.5 | 47.72+0.9 | 95.84+1.2 | 73.95+1.4 | 97.67+0.4 | 27.15+0.3 | 74.37+1.5 |
| IBOT-Aug-est+sample | 90.67+1.5 | 72.19+4.2 | 44.37+4.5 | 94.43+0.5 | 87.86+0.3 | 76.33+1.1 | 46.76+0.7 | 93.75+2.1 | 69.04+2.3 | 97.39+0.4 | 26.45+0.3 | 72.66+1.6 |
| Swin-noAug | 90.02+0.5 | 33.07+3.3 | 53.80+2.3 | 92.57+0.5 | 81.13+1.4 | 70.33+1.8 | 38.78+1.3 | 94.64+0.9 | 76.01+1.5 | 93.93+0.5 | 24.26+0.2 | 68.05+1.3 |
| Swin-Aug-est | 90.32+0.6 | 73.31+2.5 | 52.92+3.0 | 92.50+0.6 | 80.65+2.1 | 71.11+2.3 | 39.93+1.6 | 94.26+0.9 | 76.47+1.0 | 94.10+0.6 | 24.45+0.3 | 71.82+1.4 |
| Swin-Aug-est+sample | 90.56+0.5 | 69.84+3.3 | 53.03+1.7 | 91.92+0.6 | 80.96+2.0 | 71.27+1.7 | 38.86+1.3 | 92.17+1.2 | 70.74+2.0 | 93.89+0.5 | 24.67+0.2 | 70.72+1.4 |

**Whether to use augmentations for augmenting training set/estimating FIs**. When we have the opportunity to augment data, we can either use it to estimate feature importance, or augment the training set of the downstream task, or conduct both. In Table 4 and Table 5, we show the experiment results of all possibilities. We have the following observations:

(1) Our method can improve performance when no augmentations are used.

(2) Our method can improve performance when the data augmentations are used to augment the training set.

(3) Only augmenting the downstream training set, while not adjusting feature importance, *does not* lead to performance improvement under most circumstances (comparing the baseline performance of noAug and Aug-est+sample). This is particularly interesting since in many-shot training, data augmentation is always useful. This also empirically explains why in few-shot learning literature test-time data augmentations are rarely used.

The above three points highlight the fact that our method is universally useful and the improvement does not come from the data augmentation of the training set itself which may cause unfair comparison. More importantly, simply adding more data augmentations does not make improvement for few-shot transfer (as opposed to many-shot learning), and our method takes a step forward to succesfully make good use of data augmentations. We also have another observation:

(4) the performance improvement is more significant when the training set is augmented with data. However, we see from point (3) that the baseline performance without feature adjustment is often lower when data augmentation is used for augmenting the training set, thus we recommend to *only* use data augmentation for estimating feature importance.

**On the type of data augmentation.** In Table 6, we vary the type of augmention used for estimating feature importance and see the performance improvement. In addition to random cropping, we consider color jitter, random horizontal flip, and random rotation which are also commonly used data augmentation strategies (Chen et al., 2020). We also consider the combinations of random cropping and other augmentations. As seen, random cropping performs the best on average. In particular, random horizontal flip worsens the performance in 1-shot setting, meaning that using this augmentation does not even approach the real distribution; color jitter performs well on some datasets, but worsens significantly on CUB and EuroSAT. After a closer look at CUB and EuroSAT, we find that the background of images are similar across the whole CUB, and the images in EuroSAT all have similar color distribution. Thus changing color leads to out-of-distribution samples for these two datasets, leading to worse estimation when using color jitter as data augmentation. Another thing to mention is that all augmentations do not increase 1-shot performance on EuroSAT. We speculate that EuroSAT is a very special dataset that has a fixed image configuration for all images in each

Table 6: Using different data augmentations leads to different performance improvements. We use ResNet-50 trained on ImageNet as the pretrained model.

| | | CUB | Traffic | Aircraft | CropD | ESAT | Fungi | ISIC | Omniglot | QuickD | Flowers | ChestX | Avg |
|---|---|---|---|---|---|---|---|---|---|---|---|---|---|
| Crop | 1-shot | 74.93+1.6 | 54.58+2.7 | 40.74+1.9 | 74.39+2.5 | 69.52-1.0 | 50.36+2.7 | 29.58+1.1 | 80.63+1.6 | 50.96+1.2 | 79.17+3.5 | 21.85+0.2 | 56.97+1.6 |
| | 5-shot | 90.76+1.2 | 73.68+4.1 | 57.37+3.7 | 89.48+1.8 | 84.71+0.8 | 71.47+4.0 | 40.56+2.3 | 91.45+2.1 | 68.85+2.7 | 93.57+1.6 | 24.75+0.1 | 71.51+2.2 |
| Color | 1-shot | 74.36-1.1 | 54.45+3.5 | 40.87+2.2 | 74.39+1.8 | 69.54-2.0 | 50.54+0.7 | 29.39+0.4 | 80.41+0.7 | 51.03+1.4 | 79.10+2.7 | 22.00+0.0 | 56.92+0.9 |
| | 5-shot | 91.18+1.0 | 73.25+4.6 | 57.51+4.8 | 90.75+1.3 | 84.74+0.2 | 71.52+3.1 | 39.60+1.5 | 92.60+1.5 | 69.44+2.8 | 93.98+1.2 | 24.70-0.1 | 71.75+2.0 |
| Flip | 1-shot | 74.47-5.3 | 54.65-1.9 | 40.61-0.6 | 74.49-4.4 | 69.38-10.1 | 50.36-3.2 | 29.58-1.6 | 80.39-5.5 | 51.17-2.4 | 79.28-3.8 | 21.85-0.2 | 56.93-3.5 |
| | 5-shot | 91.36+1.1 | 73.09+4.6 | 58.06+4.8 | 90.68+1.5 | 84.73+0.3 | 71.64+3.4 | 39.98+1.7 | 92.71+1.6 | 69.93+2.9 | 94.19+1.2 | 24.65-0.2 | 71.91+2.1 |
| Rotate | 1-shot | 74.61+1.7 | 54.93+2.6 | 40.21+1.1 | 74.26+2.5 | 69.63-0.5 | 50.50+2.6 | 29.21+1.4 | 80.23+0.7 | 50.91+1.2 | 79.05+3.3 | 21.89+0.0 | 56.86+1.5 |
| | 5-shot | 91.22+1.2 | 73.17+3.8 | 57.92+2.8 | 90.56+1.3 | 84.87+0.4 | 71.54+4.0 | 39.66+2.3 | 92.69+0.5 | 69.70+1.4 | 93.96+1.3 | 24.62+0.1 | 71.81+1.7 |
| Crop+Color | 1-shot | 74.46+0.8 | 54.52+3.4 | 40.86+2.0 | 74.52+2.1 | 69.49-1.9 | 50.65+1.8 | 29.20+1.1 | 80.30+1.4 | 50.93+1.2 | 79.33+3.3 | 21.75+0.1 | 56.91+1.4 |
| | 5-shot | 91.23+1.2 | 73.19+4.6 | 57.79+3.9 | 90.68+1.3 | 84.89+0.1 | 71.47+3.6 | 39.61+2.0 | 92.59+1.3 | 69.60+2.1 | 94.23+1.3 | 24.60+0.0 | 71.81+2.0 |
| Crop+Flip | 1-shot | 74.66+1.5 | 54.40+2.8 | 40.55+1.9 | 74.63+2.5 | 69.24-0.9 | 50.86+2.7 | 29.35+1.2 | 80.12+1.7 | 51.09+1.1 | 79.26+3.5 | 21.86+0.0 | 56.91+1.6 |
| | 5-shot | 91.15+1.5 | 73.32+4.4 | 57.65+3.9 | 90.70+1.5 | 84.89+0.3 | 71.59+4.1 | 39.48+2.3 | 92.47+1.9 | 69.02+1.9 | 94.07+1.5 | 24.53+0.2 | 71.79+2.1 |
| Crop+Rotate | 1-shot | 74.49+1.4 | 54.23+2.6 | 40.33+1.3 | 74.35+2.5 | 69.59-0.3 | 50.44+3.0 | 29.36+1.7 | 80.45+1.1 | 50.95+1.2 | 79.13+3.6 | 21.89+0.1 | 56.84+1.6 |
| | 5-shot | 91.16+1.2 | 73.05+3.8 | 58.17+2.6 | 90.63+1.3 | 84.90+0.5 | 71.79+4.2 | 39.89+2.6 | 92.62+0.8 | 69.64+1.3 | 94.09+1.3 | 24.45+0.1 | 71.85+1.8 |

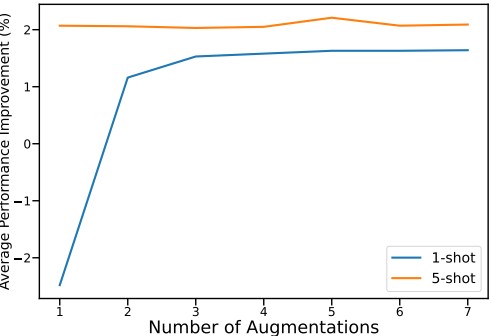

Figure 10: How the improvement brought by the proposed method changes with the number of data augmentations used. The backbone is ImageNet-pretrained ResNet-50, and the performance is averaged over all 11 downstream datasets.

class, thus all data augmentations lead to somewhat OOD samples. Overall, we see that random cropping is the most robust augmentation that works best for most scenarios.

**Varying the number of augmentations per image.** In Figure 10, we vary the number of data augmentations used for estimating feature importance and see how performance improvement changes. As seen, for 1-shot task, the performance improvement first increases, and then the performance becomes stable, while for 5-shot task, the performance improvement hangs on 2% for all numbers of augmentations. This is reasonable since the estimated data distribution of 1-shot task only relies on the data augmentation, thus we need more data augmentations to make the estimation more accurate.

**Varying the number of ways and shots.** In Table 7 and Table 8, we report the performance of our proposed method with higher ways and shots. As expected, when increasing the number of ways and shots, the performance improvement brought by the method becomes smaller due to weaker feature redundancy. However, the improvement still exists, demonstrating the broad usage of our method.

## G  ESTIMATING FEATURE IMPORTANCE FOR FINETUNE

All of the contents we discussed so far are based on the assumption that the pretrained model is frozen when transferring to downstream tasks. As shown in Luo et al. (2023), finetune generally performs better than linear probing, especially when the distribution shift is large between pretraining dataset and the downstream dataset. As finetuning the feature extractor makes changes to the features themselves, the analysis of features can be more difficult and we leave it for future work. In this section, we simply want to know whether our proposed method can be used to improve finetune performance.

Table 7: Performance improvement brought by the proposed method under different number of ways. We use ResNet-50 trained on ImageNet as the pretrained model.

| | | CUB | Traffic | Aircraft | CropD | ESAT | Fungi | ISIC | Omniglot | QuickD | Flowers | ChestX |
|---|---|---|---|---|---|---|---|---|---|---|---|---|
| 10-way | 1-shot | 62.92+2.4 | 40.01+2.6 | 27.86+1.6 | 64.35+2.3 | 56.54-0.5 | 37.58+2.7 | - | 70.03+1.7 | 37.62+1.5 | 70.37+4.0 | - |
| | 5-shot | 84.72+2.0 | 59.03+5.0 | 44.13+3.8 | 85.01+1.5 | 75.92+0.9 | 59.89+4.4 | - | 87.20+1.7 | 56.84+2.6 | 90.25+1.8 | - |
| 20-way | 1-shot | 51.22+2.7 | 28.64+2.1 | 19.30+1.2 | 54.94+1.8 | - | 28.19+2.2 | - | 60.25+1.6 | 26.81+1.4 | 61.49+4.1 | - |
| | 5-shot | 76.67+2.6 | 46.56+4.9 | 32.81+3.5 | 78.09+1.6 | - | 48.88+4.3 | - | 81.01+2.0 | 45.35+2.5 | 85.05+2.5 | - |
| 50-way | 1-shot | 37.35+2.4 | - | 12.08+1.0 | - | - | 18.91+1.6 | - | 44.21+3.4 | 17.54+1.4 | 50.91+4.0 | - |
| | 5-shot | 63.59+3.3 | - | 21.79+2.8 | - | - | 36.42+3.7 | - | 60.66+7.3 | 28.31+3.7 | 77.84+3.1 | - |
| 100-way | 1-shot | 28.49+2.1 | - | 8.80+0.6 | - | - | 13.88+1.2 | - | 37.95+1.2 | 12.08+0.9 | 43.22+3.8 | - |
| | 5-shot | 53.24+3.6 | - | 15.64+2.3 | - | - | 28.61+3.1 | - | 55.79+2.4 | 24.40+2.0 | 71.52+3.6 | - |

Table 8: Performance improvement brought by the proposed method under different number of shots. We use ResNet-50 trained on ImageNet as the pretrained model.

| | CUB | Traffic | Aircraft | CropD | ESAT | Fungi | ISIC | Omniglot | QuickD | Flowers | ChestX |
|---|---|---|---|---|---|---|---|---|---|---|---|
| 5-shot | 90.14+2.5 | 72.21+5.4 | 59.24+4.3 | 91.50+1.1 | 85.11+0.2 | 71.71+4.3 | 40.94+1.4 | 92.76+1.6 | 68.60+2.9 | 94.41+1.1 | 25.43+0.0 |
| 10-shot | 93.11+0.9 | 79.40+3.9 | 63.58+4.0 | 92.51+1.5 | 87.83+1.0 | 77.93+3.4 | 45.87+2.6 | - | 73.48+2.9 | 95.70+1.1 | 26.47+0.3 |
| 20-shot | 94.49+0.7 | 84.06+3.2 | - | 94.40+1.2 | 89.98+0.9 | 82.53+2.5 | 50.75+2.8 | - | 76.74+2.9 | 96.86+0.9 | 28.71+0.3 |
| 50-shot | - | 88.81+2.3 | - | 96.30+0.7 | 91.99+0.6 | 86.05+1.6 | 56.92+2.6 | - | 80.72+2.6 | 97.20+0.7 | 31.74+0.8 |

Before the finetune process starts, we estimate feature importance using the training set, and adjust the features with the estimated feature importance throughout the finetune process. Following Luo et al. (2023), we separate the learning rates for the backbone and the linear head. We tune the number of epochs and the learning rates with respect to the basic finetune performance (without feature adjustment) on Aircraft separately for 1-shot and 5-shot settings, and use them for all datasets. The resultant hyperparameters (epoch, backbone lr, head lr) for ImageNet-pretrained ResNet-50 are: (15, 0.002, 0.5) for 1-shot and (20, 0.01, 0.5) for 5-shot. We use SGD with momentum as the optimizer, and use cosine annealing LR schedular without restart.

The results are shown in Table 9. As can be seen, performance obtained by finetuning the whole backbone is higher than that obtained by linear probing, and our method can also improve finetuning performance, showing the universal utility of our method. The results are somehow surprising since the estimated feature importance is estimated for the features before finetuning, so intuitively it is only useful at the beginning of the finetuning process. We speculate that the estimated feature importance at the beginning serves as a prior knowledge to better guide the finetuning process, and since the data is scarce, such initial guidance is very important.

Table 9: Performance improvement brought by the proposed method when using finetune as the downstream method. We use ResNet-50 trained on ImageNet as the pretrained model.

| | CUB | Traffic | Aircraft | CropD | ESAT | Fungi | ISIC | Omniglot | QuickD | Flowers | ChestX | Avg |
|---|---|---|---|---|---|---|---|---|---|---|---|---|
| 1-shot | 76.48+1.6 | 55.33+2.9 | 43.62+0.5 | 79.21+0.8 | 72.43-0.9 | 50.93+1.6 | 34.57+0.8 | 81.11+1.3 | 54.59+0.2 | 83.13+1.0 | 22.57+0.0 | 59.45+0.9 |
| 5-shot | 91.89+1.1 | 81.11+3.4 | 59.83+4.7 | 92.20+1.1 | 87.69+0.3 | 74.63+2.9 | 53.23+3.7 | 95.15+2.1 | 73.57+2.2 | 95.83+0.8 | 24.76+0.2 | 75.44+2.0 |

