# OpenReview forum: "Less is More: On the Feature Redundancy of Pretrained Models When Transferring to Few-shot Tasks"
_ICLR.cc/2024/Conference — Submitted to ICLR 2024_

### Official Review · Reviewer_Xfoy · 2023-10-29

**Soundness:** 3 good
**Presentation:** 3 good
**Contribution:** 2 fair
**Rating:** 5
**Confidence:** 5

**Summary:**

This work find a novel phenomenon for linear probing that the representation from large pre-trained model has dimension-redundancy problem for downstream few-shot tasks. Specificly, most dimensions are redundant only under few-shot settings and gradually become useful when the number of shots increases. The authors provide a theoritical analysis on toy experiments and use feature importance estimate to select the important dimensions for each few-shot task. The experiments on many datasets and pre-trained model verify this phenomenon and the effectiveness of the proposed method.

**Strengths:**

1. This work finds an interesting phenomenon about few-shot transfer and provides detailed analysis.
2. The paper is organized and written well, and the motivation is clear to me.
3. The toy experiments and theoretical analysis are meaningful and help to understand this work.
4.The authors conduct extensive experments on many different datasets and different pre-trained models.

**Weaknesses:**

1. The proposed method is over-simple and more like a trick. And thus there are no ablation study. I suggest the method can be more systematic.
2. Fine-tuning is typically a more preferred method and achieve better performance than linear probing, so it has a weak value to solve linear probling alone. How the proposed method perform for fine-tuning?
2. As shown in Figure 2, the proposed phenomenon seems to disappear when way is large. In fact, I think the large way is more realistic and ractically useful.
3. How the proposed method perform under many-way classification? The authors only provide the results on 5-way setting, how about the results on 20-way, 50-way setting?

**Questions:**

Please respond to the weakness above.

---

> ### Author Response · Authors · 2023-11-16
> **Response to Reviewer Xfoy**
>
> Thank you for the constructive comments. We present our response to your comments below.
>
>
> **1. The proposed method is over-simple and more like a trick. And thus there are no ablation study. I suggest the method can be more systematic.**
>
> Thanks for your helpful advice! We follow your suggestions and conduct several ablation studies, including the use of augmentations for augmenting training set and estimating FIs, the type of data augmentations, the number of augmentations per image, the number of ways, and the number of shots. All these experiments have been added as Appendix F in the revised paper. As seen from the results, our method can give robust performance improvement in different settings. We hope these added contents will make our method more systematic, in order to let readers know how to follow the specific design details.
>
>
> **2. Fine-tuning is typically a more preferred method and achieve better performance than linear probing, so it has a weak value to solve linear probling alone. How the proposed method perform for fine-tuning?**
>
> We follow your suggestions and conduct experiment with full finetuning on ImageNet-pretrained ResNet-50. Please see the details in Appendix G of the revised paper. We report the results below.
>
> 1-shot: CUB: 76.48 + 1.6 Traffic Signs: 55.33 + 2.9 Aircraft: 43.62 + 0.5 Crop Disease: 79.21 + 0.8 EuroSAT: 72.43 - 0.9 Fungi: 50.93 + 1.6 ISIC: 34.57 + 0.8 Omniglot: 81.11 + 1.3 Quick Draw: 54.59 + 0.2 VGG Flower: 83.13 + 1.0 ChestX: 22.57 + 0.0 Avg: 59.45 + 0.9
>
> 5-shot: CUB: 91.89 + 1.1 Traffic Signs: 81.11 + 3.4 Aircraft: 59.83 + 4.7 Crop Disease: 92.20 + 1.1 EuroSAT: 87.69 + 0.3 Fungi: 74.63 + 2.9 ISIC: 53.23 + 3.7 Omniglot: 95.15 + 2.1 Quick Draw: 73.57 + 2.2 VGG Flower: 95.83 + 0.8 ChestX: 24.76 + 0.2 Avg: 75.44 + 2.0
>
> As can be seen, our method can improve fine-tuning performance under few-shot transfer scenarios, and sometimes by a large margin. It is very interesting to analyze how features and their importance change during finetuning, and we leave this to future work.
>
> **3. As shown in Figure 2, the proposed phenomenon seems to disappear when way is large. In fact, I think the large way is more realistic and ractically useful.**
>
> Please take another closer look at Figure 2, and we can see that the feature redundancy phenomenon weakens but not completely disappear when way is large, different from the trend of the number of shots; for example, from the figure we observe that, 255 dimensions, only 10% of the full dimensions, are enough for obtaining the original 100-way performance, thus 90% of the feature dimensions can be seen as redundant. In addition, we see that there is always a large gap between the performance obtained by Oracle feature importance adjustment (red line) and the original performance (blue line) regardless of the number of ways, leaving room for feature adjustment.
>
> **4. How the proposed method perform under many-way classification? The authors only provide the results on 5-way setting, how about the results on 20-way, 50-way setting?**
>
> Following the previous response, we conduct experiments on higher ways as part of ablation studies in Table 7 in the revised version of the paper. Please take a look. We can see that the performance improvement still exists for high ways like 100, though not that significant for some datasets, which verifies our statement in the previous response.

---

> ### Author Response · Authors · 2023-11-20
> **Addressing concerns**
>
> Dear reviewer Xfoy,
>
> Thank you again for taking the time to review our submission. As the end of the author-reviewer discussion phase is approaching, we would appreciate it if you let us know whether the responses and the changes to the paper have addressed your concerns. In any case, we remain available to answer any further questions.

---

### Official Review · Reviewer_BvhJ · 2023-10-30

**Soundness:** 3 good
**Presentation:** 3 good
**Contribution:** 3 good
**Rating:** 6
**Confidence:** 4

**Summary:**

This paper primarily addresses the issue of feature redundancy in pretrained models when transferring to downstream few-shot tasks. It then proceeds to conduct a thorough theoretical analysis to explain why this phenomenon specifically occurs in few-shot settings. Additionally, this paper introduces the concept of a soft mask based on feature importance to identify redundant dimensions. According to experimental results, this approach performs effectively in few-shot transfer scenarios.

**Strengths:**

+ This paper introduces a novel and interesting perspective suggesting that a select few feature dimensions can achieve performance comparable to using all dimensions in few-shot settings. Such a phenomenon of feature redundancy warrants deeper investigation.
+ Experimental results show that employing a soft mask based on feature importance is effective in most few-shot transfer scenarios. Additional examples in the appendix further underscore the efficacy of this method.
+ In-depth theoretical insights elucidate the reasons behind the existence of the feature redundancy phenomenon in few-shot settings. Figures like Fig 1 explicitly corroborate the idea that increasing the number of retained dimensions can initially enhance performance, then subsequently leads to a decline.

**Weaknesses:**

Identifying redundant dimensions more effectively with limited samples could be a potential avenue for future research. While this paper suggests estimating feature importance using augmented data, I have reservations about whether augmented data can genuinely replicate the true data distribution.If the data augmentation technique were altered, would it obviously influence the experimental outcomes?

As noted in the paper, the optimal data augmentation technique can vary significantly across different datasets. Is the cropping operation specifically recommended, or can other methods yield equally effective experimental results?

**Questions:**

Please refer to the weaknesses part.

---

> ### Author Response · Authors · 2023-11-16
> **Response to Reviewer BvhJ**
>
> Thank you for your positive and constructive comments. We present our response to your comments below.
>
> - **If the data augmentation technique were altered, would it obviously influence the experimental outcomes? Is the cropping operation specifically recommended, or can other methods yield equally effective experimental results?**
>
> To answer your question, we conduct additional experiments with other 3 commonly used data augmentations, including random color jitter, random horizontal flip, and random rotation. We report the average accuracy improvement over 11 datasets below. See details in Table 6 of the revised version of the paper.
>
> 1-shot: Crop: +1.6  Color: +0.9 Flip: -3.5 Rotate: +1.5 Crop&Color: +1.4 Crop&Flip: +1.6  Crop&Rotate: +1.6
>
> 5-shot: Crop: +2.2  Color: +2.0 Flip: +2.1 Rotate: +1.7 Crop&Color: +2.0 Crop&Flip: +2.1  Crop&Rotate: +1.8
>
> Thus on average, cropping performs the best. We particularly note that color jitter performs well on almost all datasets, except for CUB where the performance degrades for 1-shot. We note that all images of each class in CUB (one particular bird) share a similar background with consistent color, thus we speculate that changing color distribution will produce OOD samples for CUB. This verifies that a good augmentation can be good on some datasets and performs badly on another. However, we expect that random cropping, by design, may serve as a good augmentation under most scenarios (in our experiments on a lot of pretrained models, it performs not well only on the single 1-shot case of ImageNet-pretrained ResNet-50 on EuroSAT). We do not think that data augmentations can really replicate the real data distribution, but they are good enough to serve as an approximate surrogate (better than nothing) to improve performance.

---

> ### Author Response · Authors · 2023-11-20
> **Addressing concerns**
>
> Dear reviewer BvhJ,
>
> Thank you again for taking the time to review our submission. As the end of the author-reviewer discussion phase is approaching, we would appreciate it if you let us know whether the responses and the changes to the paper have addressed your concerns. In any case, we remain available to answer any further questions.

---

> > ### Comment · Reviewer_BvhJ · 2023-11-22
> >
> > Thank you for your feedback and the additional experiments, which have effectively addressed my concerns. Consequently, I will maintain my original score.

---

### Official Review · Reviewer_YUQa · 2023-11-02

**Soundness:** 2 fair
**Presentation:** 2 fair
**Contribution:** 2 fair
**Rating:** 3
**Confidence:** 4

**Summary:**

This paper highlights feature redundancy in transferring pretrained models to few-shot tasks using linear probing protocols. The study also offers theoretical analysis and proposes a data augmentation-based solution.

**Strengths:**

The paper provides a thorough evaluation of various downstream tasks using multiple pretrained models, both supervised and self-supervised.

The paper is well-written.

**Weaknesses:**

The term *feature redundancy* in the paper is somewhat misleading, as it doesn't provide evidence of mutual information between feature distributions.  It is more like *some features are distinguishable for downstream tasks, while others are not*.

The findings, from the perspective that *'some features are distinguishable'*, are not particularly surprising. It's expected that features in a pretrained model may vary in their relevance to downstream tasks, and different tasks may have distinct feature preferences. Sensitivity to linear classifiers on indistinguishable features with few samples is also a common expectation, and the improvement from data augmentation can be viewed as adding more data.

**Questions:**

Is it possible to evaluate the mutual information of features?

---

> ### Author Response · Authors · 2023-11-16
> **Response to Reviewer YUQa (Part 1/2)**
>
> Thank you for your helpful comments. We present our response to your comments below.
>
>
> **1. The term feature redundancy in the paper is somewhat misleading, as it doesn't provide evidence of mutual information between feature distributions. It is more like some features are distinguishable for downstream tasks, while others are not. Is it possible to evaluate the mutual information of features?**
>
> We agree that we should give evidence that feature dimensions share much repetitive information, in order to prove that features are indeed redundant by definition. We find that calculating mutual information among a lot of dimensions is difficult, so we instead calculate the singular values of the feature covariance matrix to see how many dimensions are repetitive and redundant, following prior work [1,2]. Please see https://app.gemoo.com/share/image-annotation/583217980207886336?codeId=Ml2plBWX0B6NY&origin=imageurlgenerator&card=583217976009388032 or Figure 1(a) of the revised paper which shows the distribution of singular values of feature covariance matrix. The features are extracted from ImageNet-pretrained ResNet-50 on Aircraft. We can see that only very few singular values are larger than zero. In fact, the effective rank of features [2,3] is 86.40, only about 4.3% of the feature dimensionality 2048. Thus the pretrained features are indeed extremely repetitive and redundant. The observations in our paper further show that those non-informative features (or possibly some informative ones that lead to even more useless features) are not only redundant but also harmful for few-shot transfer problems. With this additional experiment available, we think the term feature redundancy is now appropriate.
>
>
> [1] Jing et al. Understanding Dimensional Collapse in Contrastive Self-supervised Learning. ICLR 2022.
>
> [2] Zhuo et al. Towards a Unified Theoretical Understanding of Non-contrastive Learning via Rank Differential Mechanism. ICLR 2023.
>
> [3] Roy et al. The effective rank: A measure of effective dimensionality.
>
> **2. The findings, from the perspective that 'some features are distinguishable', are not particularly surprising. It's expected that features in a pretrained model may vary in their relevance to downstream tasks, and different tasks may have distinct feature preferences.**
>
>
> We would like to mention that in our paper we find that we can **improve several points of few-shot performance by abandoning 95% feature dimensions of pretrained models, and keep performance unchanged by abandoning 99% feature dimensions (meaning that only 1% is well-distinguishable)**. Even if one may expect that features in a pretrained model may vary in their relevance to downstream tasks, we do not think one can expect that almost all feature dimensions can be abandoned in a task without influencing performance. This is what makes our findings surpring. Also, knowing that distinguishable/important feature dimensions may vary across tasks does not imply that we can locate these dimensions accurately, and this is another important contribution of our paper. We sincerely request the reviewer list relevant papers that can support the "not particularly surprising" argument.
>
> **3. Sensitivity to linear classifiers on indistinguishable features with few samples is also a common expectation**
>
> We do not agree that if without rigorous theoretical proof, one can easily expect that linear classifiers are sensitive to unimportant/indistinguishable features with **few** samples, not many. Even if one has such expectation, turning intuition into theorems also needs more effort and has its own values. Our theorem 5.1 and 5.2 further gives an estimate of the degree of this sensitivity, which is also important. If you know any relevant papers that can support your argument "this is a common expectation", please let us know.

---

> ### Author Response · Authors · 2023-11-16
> **Response to Reviewer YUQa (Part 2/2)**
>
> **4. The improvement from data augmentation can be viewed as adding more data.**
>
> We understand your concerns that the advantage of the method may come from the data augmentation itself, not from the proposed method. To alleviate your concerns, we conduct two additional experiments: 1. do not use data augmentation (noAug) and 2. use data augmentations to augment the training set and/or estimate feature importance. We show the average 5-way 5-shot performance and the improvement brought by the method over all 11 datasets in the Table below. For details and 1-shot experiments please see Table 4 and Table 5 in the revised version of our paper. Aug-est refers to only using augmentations for estimating feature importance, while
> Aug-est+sample refers to using augmentations for augmenting training set as well.
>
> | Model     | noAug| Aug-est| Aug-est+sample |
> |-----------|---------------------|-----|---------------------|
> | DINOv2    |  77.48 + 0.9  |  77.84 + 1.0  |  76.07 + 1.2  |
> | EsViT     | 71.00 + 1.8  |  71.15 + 1.9  |  69.57 + 1.9  |
> | ResNet-50 | 72.00 + 2.3  |  71.51 + 2.2  |  70.02 + 2.5  |
> | CLIP      | 76.22 + 2.3 | 75.83 + 3.0  |  74.05 + 3.1  |
> | IBOT      | 74.14 + 1.4 | 74.37 + 1.5 | 72.66 + 1.6 |
> | Swin      | 68.05 + 1.3 | 71.82 + 1.4 | 70.72 + 1.4 |
>
> We can see from the result that (1) our method without data augmentations can still obtain significant performance improvement; (2) simply putting data augmentations into the downstream training set cannot bring improvement in most circumstances, while our method can take advantage of data augmentations. (3) our method can work when augmenting the downstream training set with data augmentation.
>
> Thus we can see that **the benefit of our method is universal regardless of the use of data augmentations, and more importantly, simply adding more data augmentations does not make an improvement for few-shot transfer (as opposed to many-shot learning), and our method takes a step forward to successfully make good use of data augmentations. Thus the improvement from data augmentation cannot be simply viewed as adding more data.** It is quite interesting to see why simple data augmentations for augmenting the training set do not work for few-shot transfer settings in the future.

---

> ### Author Response · Authors · 2023-11-20
> **Addressing concerns**
>
> Dear reviewer YUQa,
>
> Thank you again for taking the time to review our submission. As the end of the author-reviewer discussion phase is approaching, we would appreciate it if you let us know whether the responses and the changes to the paper have addressed your concerns. In any case, we remain available to answer any further questions.

---

### Official Review · Reviewer_h7zL · 2023-11-04

**Soundness:** 2 fair
**Presentation:** 3 good
**Contribution:** 2 fair
**Rating:** 8
**Confidence:** 5

**Summary:**

The paper explores the phenomenon of feature redundancy in pre-trained models when applied to few-shot learning tasks. The paper demonstrates that not all dimensions of the pre-trained features are useful for a given downstream task, especially when the data is scarce. In some cases, using only 1% of the most important feature dimensions can recover the performance achieved by using the full representation. This feature redundancy is particularly prominent in few-shot settings and diminishes as the number of samples increases.
The paper also delves into the theoretical understanding of this phenomenon, showing how dimensions with high variance and small distance between class centroids can serve as confounding factors that disturb classification results in few-shot settings. To address the issue of feature redundancy, the paper proposes adjusting feature magnitude with a soft mask based on estimated feature importance. This method is shown to generally improve few-shot transfer performance across various pre-trained models and downstream datasets.

**Strengths:**

1. This paper studies an interesting and important problem of feature redundancy, especially in the context of few-shot learning.
2. This paper provides a theoretical framework that sheds light on the complex phenomenon of feature redundancy.
3. This paper also proposes a novel solution to the problem of feature redundancy by adjusting feature magnitude with a soft mask based on estimated feature importance. This is a practical approach that shows promise in improving few-shot transfer performance.
4. The paper is well-structured and clearly written, making it accessible to both experts and those new to the field.

**Weaknesses:**

Despite the above strengths, this paper also has some drawbacks:

1. While this paper studies the few-shot problem, it mainly focuses on the nearest-centroid classifier (NCC) and employs Euclidean distance as the distance metric. However, it is worth noting that most practical few-shot methods use **cosine distance** rather than Euclidean distance. Therefore, to enhance precision and applicability, it is advisable for the paper to explicitly limit its claims to Euclidean distance-based scenarios. It would be beneficial for the authors to acknowledge this limitation and potentially explore how the theoretical and empirical analysis may adapt to cosine distance-based methods. This could provide a more comprehensive understanding of the practical implications of the work.

2. I understand that different features have different importance [1]. The central argument of this paper is that utilizing a reduced set of crucial features outperforms the use of all available features. However, I noticed another paper [2] providing empirical evidence to the contrary, suggesting that "tail" features are more important than the core features in the few-shot problem.

- I wonder if the contradiction between these perspectives comes from the use of distinct distance metrics, as [2] employs Kendall distance. If so, this reminds us again that this work is primarily applicable within the context of Euclidean distance.

3. The proposed solution relies on estimating feature importance, which itself can be challenging in a few-shot setting as mentioned in the paper. A compromise method is applying data augmentation to the training data. However, this is a little bit unfair for few-shot learning since it could be regarded as an implicit means of augmenting the number of shots through the incorporation of human knowledge.

4. The math part of this paper has many unclear areas:

- Page 6: $z$ depends on label $y$, and thus you should use different subscripts to distinguish them. For example, $z_y\sim N([\mu_{(y,1)},\mu_{(y,2)}]^T, diag(\sigma_1^2, \sigma_2^2)))$. Moreover, you should remind readers that you use the same $\sigma_1$ and $\sigma_2$ for different $y$, since they are different in Eq. (1).

- Theorem 5.1: The formula of error rate in Eq. (3) is not intuitive enough. I suggest to use the equivalent form $\Pr[|(z_1,z_2)-(p_{(a,1)},p_{(a,2)})| > |(z_1,z_2)-(p_{(b,1)},p_{(b,2)})|]$ instead.

- Theorem 5.2: The simultaneous presence of both "with probability at least 0.9 (a certain number)" and "big O" renders this theorem meaningless. For example, choosing $999999999\sqrt{\log n/n}$ as $O(\sqrt{\log n/n})$ and choosing $0.0000001\sqrt{\log n/n}$ as $O(\sqrt{\log n/n})$ should have different probabilities. I believe it is possible to derive a closed-form expression for $O(\sqrt{\log n/n})$.


---
Ref:

[1] Channel Importance Matters in Few-Shot Image Classification.

[2] DiffKendall: A Novel Approach for Few-Shot Learning with Differentiable Kendall's Rank Correlation.

**Questions:**

Why are linear probing and NCC actually the same for 1-shot task? (Table 1)

---

> ### Author Response · Authors · 2023-11-16
> **Response to Reviewer h7zL (Part 1/2)**
>
> Thank you for your constructive comments. We present our response to your comments below.
>
> **1. While this paper studies the few-shot problem, it mainly focuses on the nearest-centroid classifier (NCC) and employs Euclidean distance as the distance metric. However, it is worth noting that most practical few-shot methods use cosine distance rather than Euclidean distance.**
>
> We apologize for the misunderstanding caused by negligence in writing. We normalized all pretrained features to have unit $l_2$ norm throughout the paper but forgot to mention this point. Thus in fact, **we are using cosine distance, not Euclidean distance** in all our experiments except for the theoretical part. Also, the toy example in the theoretical part shows that using Euclidean distance can lead to feature redundancy as well. We will further show in our next point of response that using other metrics like Kendall distance also leads to the feature redundancy phenomenon. Thus the feature redundancy problem, as an intrinsic problem of pretrained features, generally exists regardless of the transfer-time classifier used. In addition, if interested, the reviewer can view Appendix G which shows that even using full finetune as the adaptation method, the feature redundancy problem still exists (as part of the response to Reviewer Xfoy). We have revised the paper to explicitly mention the use of cosine distance in Appendix C.
>
> **2. I understand that different features have different importance [1]. I noticed another paper [2] providing empirical evidence to the contrary, suggesting that "tail" features are more important than the core features in the few-shot problem.**
>
>  We would like to clarify that, the two papers are essentially **suggesting the same idea**. The confusion comes from the different definitions of feature importance: in [2], the authors define core features as the ones having the largest average magnitude, while in [1], the authors define core features using Eq. (1) shown in our paper. If we dive into paper [1] and look at Figure 6 in their paper, we will immediately realize that **core features defined by [1]** are mostly those features with a small average magnitude, which are **non-core features defined by [2]**. Thus two papers both suggest that **pretrained neural networks put wrong emphasis on feature dimensions with large average magnitude**, and our paper goes further, showing that most dimensions (non-core by definition in [1] but core by definition in [2]) carry disturbing information for few-shot transfer and should be removed entirely.
>
> **3. I wonder if the contradiction between these perspectives comes from the use of distinct distance metrics, as [2] employs Kendall distance. If so, this reminds us again that this work is primarily applicable within the context of Euclidean distance.**
>
> To further alleviate your concerns, we experiment with Kendall distance to see if we can observe the same phenomenon. Please visit https://app.gemoo.com/share/image-annotation/583219007069962240?codeId=DGVkWVjJJrR6O&origin=imageurlgenerator&card=583219004561768448 to see **exactly the same feature redundancy phenomenon with Kendall distance**. We also show below how our proposed method can improve few-shot transfer performance when using Kendall distance in the downstream task:
>
> 1-shot: CUB: 86.39 + 1.3 Traffic Signs: 62.26 + 1.6 Aircraft: 65.97 + 0.6 Crop Disease: 78.90 + 1.3 EuroSAT: 64.71 + 1.2 Fungi: 54.21 + 1.7 ISIC: 30.76 + 0.4 Omniglot: 86.73 + 1.1 Quick Draw: 64.93 + 1.4 VGG Flower: 93.75 + 0.8 ChestX: 21.19 + 0.1
>
> 5-shot: CUB: 96.74 + 0.1  Traffic Signs: 81.30 + 1.5  Aircraft: 81.28 + 0.3 Crop Disease: 92.91 + 0.4 EuroSAT: 82.58 + 0.6 Fungi: 74.99 + 1.6 ISIC: 42.30 + 0.7 Omniglot: 95.88 + 0.5 Quick Draw: 84.32 + 0.5 VGG Flower: 99.16 + 0.1 ChestX: 22.94 + 0.1
>
> In the experiment above we use the visual encoder of CLIP as the pretrained model.

---

> ### Author Response · Authors · 2023-11-16
> **Response to Reviewer h7zL (Part 2/2)**
>
> **4. Applying data augmentation to the training data is a little bit unfair for few-shot learning since it could be regarded as an implicit means of augmenting the number of shots through the incorporation of human knowledge.**
>
> We understand your concerns that the advantage of the method may come from the data augmentation itself, not from the proposed method. To alleviate your concerns, we conduct two additional experiments: 1. do not use data augmentation (noAug) and 2. use data augmentations to augment the training set and/or estimate feature importance. We show the average 5-way 5-shot performance and the improvement brought by the method over all 11 datasets in the Table below. For details and 1-shot experiments please see Table 3 and Table 4 in the revised version of our paper. Aug-est refers to only using augmentations for estimating feature importance, while Aug-est+sample refers to using augmentations for augmenting training set as well.
>
> |Model|noAug|Aug-est|Aug-est+sample|
> |-|-|-|-|
> |DINOv2|77.48 + 0.9|77.84 + 1.0|76.07 + 1.2|
> |EsViT|71.00 + 1.8|71.15 + 1.9|69.57 + 1.9|
> |ResNet-50|72.00 + 2.3|71.51 + 2.2|70.02 + 2.5|
> |CLIP|76.22 + 2.3|75.83 + 3.0|74.05 + 3.1|
> |IBOT|74.14 + 1.4|74.37 + 1.5|72.66 + 1.6|
> |Swin|68.05 + 1.3|71.82 + 1.4|70.72 + 1.4|
>
> We can see from the result that (1) our method without data augmentations can still obtain significant performance improvement; (2) simply putting data augmentations into the downstream training set cannot bring improvement in most circumstances, while our method can take advantage of data augmentations. (3) our method can work when augmenting the downstream training set with data augmentation.
>
> Thus we can see that the benefit of our method is universal regardless of the use of data augmentations, and more importantly, simply adding more data augmentations does not make an improvement for few-shot transfer (as opposed to many-shot learning), and our method takes a step forward to successfully make good use of data augmentations. We have noticed that in few-shot learning literature, data augmentations are rarely used for support data at the adaptation time (except for some generation-based methods that design specifically-desined augmentations that only work for in-domain setting), so this is in line with our findings, and it is quite interesting to see why data augmentations do not work for few-shot transfer settings in the future.
>
> **5. Page 6: z depends on label y, and thus you should use different subscripts to distinguish them. Moreover, you should remind readers that you use the same $\sigma_1$ and $\sigma_2$ for different y, since they are different in Eq. (1). Theorem 5.1: The formula of error rate in Eq. (3) is not intuitive enough. I suggest to use the equivalent form instead.**
>
> Thanks for pointing out these typos and giving constructive suggestions. We have revised the paper according to your advice.
>
> **6. Theorem 5.2: The simultaneous presence of both "with probability at least 0.9 (a certain number)" and "big O" renders this theorem meaningless.**
>
> We would like to clarify that the utilization of big O and "with probability at least 0.9" is to simplify the presentation of our theoretical results. This is because the goal of Theorem 5.2 is to show adding new dimensions is statistically useful for improving the performance, the additional term $O(\sqrt{\log(n)/n})$ comes from the uncertainty characterization of the random draw of training set, which can be vanishing when increasing the training sample size.
>
> Additionally, the constant in the big-O notation can be certainly obtained by conducting the proof in a more precise way. For instance, **in our proof, the constant in the Hoeffding's inequality can be exactly obtained to be $1/\sqrt{2}$, and the covering number of the Hypothesis can be also upper bounded by some detailed configurations of the hypothesis set. We have modified our proof for Theorem 5.2 in the appendix of the revised version,  which shows that the constant in the Big-O notation can be roughly bounded by 9.** Thus our bound is definitely meaningful. We further emphasize that such a constant is proved by combining many loose inequalities and can be certainly improved with more detailed calculations, but this is never the scope of this paper.
>
> **7. Why are linear probing and NCC actually the same for 1-shot task?**
>
> We here consider that linear probing, without prior knowledge, will give a no-class-bias solution to the 1-shot task with propoer regulations. In fact, since for 1-shot task, there is no other information in addition to the single data for each class, a **good** linear classifier with the best performance expectation should place the classification boundary in the middle of data points, perpendicular to the line connecting them, which is exactly the same with NCC. We empirically find that minimizing Logistic Regression loss leads to this solution (using Sklearn library). We include this clarification in Appendix C.

---

> > ### Comment · Reviewer_h7zL · 2023-11-19
> > **Thanks for the Detailed Rebuttal**
> >
> > Thank you for your comprehensive and detailed rebuttal. I have carefully read through your responses to the comments and truly appreciate the time and effort you have put into addressing each concern.
> >
> > I am satisfied with the authors' responses to my original questions Q2, Q3 and Q5.
> > If the authors can **incorporate the following revision into the paper** to enhance its completeness, I am willing to increase my score.
> > 1. Regarding the empirical part, I suggest including experiments with different distances, namely, cosine distance, **not normalized Euclidean distance**, and **Kendall distance**. This addition would significantly broaden the scope of application. For example, the additional experiments shown in (https://app.gemoo.com/share/image-annotation/583219007069962240?codeId=DGVkWVjJJrR6O&origin=imageurlgenerator&card=583219004561768448).
> > 2. In the theory part, I observed that the covering number is still expressed in big O between Eq (19) and (20). Please **carefully review** the proof for precision.
> > 3. Please **replace the big O notation with the constant in the main paper** correspondingly to make the theorem more precise. For the sake of rigor, it is essential for the theorem to be precise and self-contained. Additionally, I noticed that the theorem holds only when $n>4$ as stated before the last equation of the proof. Please add the condition to the theorem. Furthermore, since the LHS of Thm 5.2 is at most one, there should be a condition for $9\sqrt{\frac{\log n}{n}}<1$ to avoid trivializing the theorem. Please **check the condition and add it to the theorem**.
> > 4. Please **add the discussion of the relationship among [1], [2] and this paper**, especially the argument that core features defined by [1] are mostly those features with a small average magnitude, which are non-core features defined by [2], and this paper further shows that most dimensions (non-core by definition in [1] but core by definition in [2]) carry disturbing information for few-shot transfer and should be removed entirely.

---

> > > ### Author Response · Authors · 2023-11-20
> > >
> > > Thanks for your quick response! We are pleased to see that most of your concerns have been addressed, and thanks for your actionable suggestions that can help us improve the paper. We have modified the manuscript following your suggestions:
> > >
> > > - We have incorporated experiments for NCC with all three different distances in Appendix E (Figure 7 and Table 3). We can see that using unnormalized Euclidean distance indeed leads to feature redundancy as well.
> > > - We have corrected the big O typo in the proof. Thanks for pointing this out. We have checked the proofs and there are no more typos.
> > > - We have replaced the big O notation with the constant in the main paper correspondingly, and added all conditions to the theorem.
> > > - We add the discussion of the relationship among [1], [2], and this paper in an additional paragraph in the related work.

---

> > > > ### Comment · Reviewer_h7zL · 2023-11-20
> > > >
> > > > Dear authors,
> > > >
> > > > I appreciate the effort you've put into addressing my concerns, which has significantly clarified my understanding of your work. Your responses have demonstrated a strong commitment to improving the paper and have alleviated most of my initial reservations. As a result, I am inclined to raise my score and recommend acceptance of your submission. I look forward to the potential impact your research could have on the ICLR community.

---

> > > > > ### Author Response · Authors · 2023-11-20
> > > > >
> > > > > Thanks for raising the score to an acceptance level. We sincerely appreciate your valuable feedback. Your constructive review and active engagement in the discussion have been instrumental in enhancing the quality of the paper.

---

### Author Response · Authors · 2023-11-16
**Revised manuscript uploaded**

We thank all the reviewers for their helpful feedback and suggestions. We have responded to their comments individually. We have also uploaded an updated version of the manuscript incorporating their suggestions and some additional experiment results. We provide a brief list of manuscript changes and additional experiments below.

1. To respond to Reviewer Xfoy's request and alleviate some reviewers' concerns about our proposed method, we add ablation studies on our proposed method in Appendix F, including the use of augmentations for augmenting training set and estimating FIs (Reviewer h7zL and YUQa), the type of data augmentations (Reviewer BvhJ), the number of augmentations per image, the number of ways (Reviewer Xfoy), and the number of shots. All results demonstrate the robustness of our methods in different settings.
2. To further prove that the features are indeed repetitive and redundant by definition, as requested by Reviewer YUQa, we add an additional Figure 1(a) depicting the singular value spectrum of the feature space. We can see from the figure that most singular values are extremely small, verifying that the features are indeed highly relevant and are thus redundant.
3. As requested by Reviewer Xfoy, we apply our method to full-finetuning and give results in Appendix G. The results show that our method can improve finetuning performance under few-shot settings.
4. We have corrected the typos noticed by Reviewer h7zL and revised the manuscripts following Reviewer h7zL's suggestions.
5. We added more evaluation details in Appendix C to avoid potential misunderstandings, e.g., the use of cosine distance mentioned by Reviewer h7zL.
6. We modified the proof of Theorem 5.2 to explicitly figure out the constant in the big-O notation, as requested by Reviewer h7zL.

We hope that our responses have addressed the reviewer's concerns, and look forward to further discussions.

---

### Meta-Review · Area_Chair_fW48 · 2023-12-08

**Metareview:**

The submission investigates the redundancy of pretrained features when transferring from a source task to a target task via linear probing. It presents empirical observations and theoretical results supporting the assertion that most pretrained features are redundant in the few-shot data regime and that this phenomenon disappears as the dataset size increases. The submission also demonstrates that while estimating feature importance is difficult in the few-shot regime, it can be used to construct a soft mask on pretrained features which helps improve few-shot classification performance.

Reviewers feel positive about the submission's interest to the research community (h7zL, BvhJ, Xfoy), its thorough evaluation (YUQa, Xfoy) and the promising results it presents (h7zL, BvhJ, Xfoy). They also note the paper's clarity and writing quality. The authors address many of the concerns raised by reviewers in their response, and for brevity I will outline the ones which I don't consider fully resolved:

- Reviewers h7zL is concerned that data augmentations could be implicitly adding more "shots" and make the comparison to other few-shot learning approaches unfair, and Reviewer YUQa is concerned that the data augmentations alone could be responsible for the improvements observed. The authors respond by reporting results which show that their proposed approach still yields improvements in the absence of data augmentations. This satisfies Reviewer h7zL, but Reviewer YUQa objects that using data augmentation to compute feature importance for selecting distinguishable features is not a novel idea (Luo et al., 2022) and that this takes away from the submission's contributions.
- Reviewer YUQa pushes back against the use of the term "feature redundancy", as the paper does not provide evidence of mutual information between feature distributions. They consider the phenomenon studied by the submission more akin to the fact that "some features are distinguishable for downstream tasks, while others are not", which to them is not a surprising claim. The authors ask for pointers to prior works studying the phenomenon and present results on singular values of the feature covariance matrix which they claim shows that pretrained features are extremely repetitive and redundant. This does not convince the reviewer: "While these features remain inactive when applied to airplane images (resulting in small covariance values), [...] this lack of activation does not imply redundancy but rather a failure to distinguish between the features in the context of airplane images. The distinction between redundancy and non-distinguishability is crucial, and the paper lacks convincing evidence to establish redundancy while excluding the possibility of non-distinguishability. Therefore, the use of the term 'redundancy' may be inaccurate or requires additional supporting evidence." The reviewer remains convinced that non-distinguishability is not a surprising property.

Given the above, and given the overall lack of a strong support from a majority of reviewers, the submission does not quite meet the bar for acceptance.

**Justification For Why Not Higher Score:**

The submission lacks strong support from a majority of reviewers and does not quite meet the bar for acceptance.

**Justification For Why Not Lower Score:**

N/A

---

### Decision · Program_Chairs · 2024-01-16

Reject